

# East Asian dust storm in May 2017: observations, modelling and its influence on Asia-Pacific region

Xiao-Xiao Zhang[1, 2], Brenton Sharratt[3], Lian-You Liu[4], Zi-Fa Wang[2], Xiao-Le Pan[2], Jia-Qiang Lei[1], Shi-Xin Wu[1], Shuang-Yan Huang[1], Yu-Hong Guo[2], Jie Li[2], Xiao Tang[2], Ting Yang[2], Yu Tian[2], Xue-Shun Chen[2], Jian-Qi Hao[2], Hai-Tao Zheng[2], Yan-Yan Yang[4], and Yan-Li Lyu[4]

[1]State Key Laboratory of Desert and Oasis Ecology, Xinjiang Institute of Ecology and Geography, Chinese Academy of Sciences, Urumqi, 830011, China
[2]State Key Laboratory of Atmospheric Boundary Layer Physics and Atmospheric Chemistry, Institute of Atmospheric Physics, Chinese Academy of Sciences, Beijing, 100029, China
[3]USDA-ARS, 215 Johnson Hall, Washington State University, Pullman, WA 99164, USA
[4]Key Laboratory of Environmental Change and Natural Disaster, Ministry of Education, Beijing Normal University, Beijing, 100875, China

*Correspondence to*: L.Y. Liu (lyliu@bnu.edu.cn) and Z.F. Wang (zifawang@mail.iap.ac.cn)

**Abstract.** A severe dust storm event originated from the Gobi Desert in Central and East Asia during 2-7 May, 2017. Based on moderate resolution imaging spectroradiometer (MODIS) satellite products, hourly environmental monitoring measurements from 367 Chinese cities and more than 2000 East Asian meteorological observation stations, and numerical simulations, we analysed the spatial and temporal characteristics of this dust event as well as its associated impact on the Asia-Pacific region. The maximum observed hourly $PM_{10}$ (particulate matter with an aerodynamic diameter $\leq$10 μm) concentration was above 1000 μg m$^{-3}$ in Beijing, Tianjin, Shijiazhuang, Baoding, and Langfang and above 2000 μg m$^{-3}$ in Erdos, Hohhot, Baotou, and Alxa in northern China. This dust event affected over 8.35 million km$^2$, or 87% of mainland China, and significantly deteriorated air quality in 316 cities of the 367 cities examined across China. The maximum surface wind speed during the dust storm was 23-24 m s$^{-1}$ in the Mongolian Gobi Desert and 20-22 m s$^{-1}$ in central Inner Mongolia, indicating the potential source regions of this dust event. Lidar-derived vertical dust profiles in Beijing, Seoul, and Tokyo indicated dust aerosols were uplifted to an altitude of 1.5-3.5 km whereas simulations by the Weather Research and Forecasting with Chemistry (WRF-Chem) model indicated 20.4 Tg and 5.3 Tg of aeolian dust being deposited respectively across continental Asia and the North Pacific Ocean. According to forward trajectory analysis by the FLEXible PARTicle dispersion (FLEXPART) model, the East Asian dust plume moved across the North Pacific within a week. Dust concentrations decreased from East Asian continent across the Pacific Ocean from a magnitude of 10$^3$ to 10$^{-5}$ μg m$^{-3}$, while dust deposition intensity ranged from 10$^4$ to 10$^{-1}$ mg m$^{-2}$. This dust event was unusual due to its impact on continental China, Korea, Japan and North Pacific Ocean. Asian dust storms such as observed in early May 2017 may lead to wider climate forcing on a global scale.

**Key words:** Air quality; Climate change; East Asia; Gobi Desert; Particulate matter



## 1 Introduction

A major dust storm arose and swept over East Asia on 2-7 May, 2017. This dust storm originated from the deserts of Central and East Asia, namely the Mongolian Gobi Desert, Taklimakan Desert, Hexi Corridor, and Alxa Desert (Fig. 1). Visibility was reduced to <100 m as a result of dense dust near Guaizihu (41.37°N, 102.37°E) on the north edge of the Badain Jaran

sand desert and Turpan (42.83°N, 89.25°E) on the north edge of Taklimkan Desert while maximum surface wind speed reached 24 m s$^{-1}$ in the Mongolian Gobi Desert (43.1°N, 109.2°E). Air quality was consequently very poor during this dust storm. This storm was unusual in that East Asian dust storms usually develop along more south-eastward trajectories (Satake et al., 2004; Nee et al, 2007; Huang et al., 2008). Dust storms develop as a result of passage of synoptic cold fronts (Hsu et al., 2006). Over the past five decades, the frequency of Asian dust events has decreased due to ecological restoration efforts

and climate change (Shao, 2011a; Lyu et al., 2017a). Strong dust storms have not occurred in the past ten years. Although East Asian dust storms mostly occur during springtime (Zhang et al., 1997; Shao and Dong, 2006; Chen, 2010; Li et al., 2012), it is essential to determine dust sources, emission, transport, and deposition to enhance our understanding of the negative impact of atmospheric dust on global warming (Tegen et al., 1996; Stanhill, 2005; Park et al., 2011; IPCC, 2013; Carslaw et al., 2013; Huang et al., 2014).

Dust aerosols can be transported long distances, even on a global scale (Merrill et al., 1994; Uno et al., 2009; Shao et al., 2011a). Atmospheric dust has been observed across continents and oceans, giving rise to its importance in both terrestrial and marine ecosystems (Huebert et al, 2003; Mahowald et al., 2009; IPCC, 2013; Kok et al., 2017). Mineral aerosols can influence air quality by reducing visibility and bolstering concentrations of inhalable particulate matter (Sharratt et al., 2006; Huneeus et al., 2011; Goudie, 2014). In dust source regions, atmospheric dust concentrations can approach 1-10 mg m$^{-3}$

while dust particles can be uplifted to altitudes of 2-10 km under strong wind shear stress (McTainsh and Strong, 2007; Cottle et al., 2013; Shao, 2013; Goudie, 2014). Fine particulates reaching high altitudes may be transported by tropospheric winds (Shao, 2000; Eguchi et al., 2009). Trans-Pacific transport of mineral dust from East Asia to North America has been frequently detected during springtime (Merrill et al., 1989; Mahowald et al., 1999; Uno et al., 2001; Zhao et al., 2003; Gong et al., 2006; McKendry et al., 2008; Cottle et al., 2013). Long-range transport of dust aerosol has been substantiated by ice

core samples taken at the North Pole (Jaffe et al., 1999; Mahowald et al., 2017). In addition, oceanic chlorophyll observations have suggested a strong linkage between mineral dust and oceanic primary production after atmospheric dust deposition events (Young et al., 1991; Mahowald et al., 2009). Duce et al. (1991) and Tagliabue et al. (2017) also suggest a possible association between dust storms and enrichment of iron (Fe) and phosphorus (P) in oceans. Mineral dust aerosols are the primary source of Fe in the atmosphere (Mahowald et al., 2017). According to physically-based model estimation,

more than 60 Tg Fe and 1 Tg P are deposited into world oceans as a result of fallout of atmospheric dust (Luo et al., 2005; Mahowald et al., 2017). However, uncertainties exist in assessing the global dust cycle due to a lack of understanding in parameterizing dust emission, transport, and deposition (Wang et al., 2000; Ginoux et al., 2001; Gong et al., 2003; Mahowald et al., 2003; Shao, 2006; Zhao et al., 2010; Huneeus et al., 2011; Kok, 2011). Therefore, the estimation of the





impact of dust on the biogeochemical cycle is still uncertain and unclear (Takemura et al., 2002; Mahowald et al., 2005; Mahowald et al., 2009; Field et al., 2010; Huneeus et al., 2011; Shao et al., 2013).

Asian dust has long been an environmental concern to China, which notably has affected the formation of the Loess Plateau and historic Chinese civilization (Tsoar and Pye, 1987; An et al., 1991; Zhang et al., 1996; An, 2000; Husar et al., 2001;

Chen et al., 2014; Goudie, 2014; Huang et al., 2014). To quantify and assess the impact of dust cycles on the environment, remote sensing (i.e. satellite) and modelling techniques based on physical processes must be used along with environmental monitoring data and surface observations (Shao et al., 2011a). The purpose of this study was therefore to determine dust emission, transport, and deposition during the May 2017 Asian dust storm using environmental observations and remote sensing data along with simulation techniques. This combined approach to understanding the fate of windblown dust will aid

in identifying the range of transport of dust across East Asia and the North Pacific Ocean.

## 2 Materials and methods

### 2.1 Data sources

2.1.1 Environmental monitoring data

Environmental monitoring data were collected in mainland China during the 2-7 May, 2017 dust storm event. $PM_{10}$ and

$PM_{2.5}$ concentrations were measured regularly at environmental monitoring stations maintained by the Ministry of Environmental Protection (MEP), China. Data collected at 367 stations were used in this study and represent a spatiotemporal distribution across continental China (Fig. 2). Ambient $PM_{10}$ and $PM_{2.5}$ concentrations were measured with an automatic beta radiation attenuation monitor designed to continuously collect particulate matter and which is widely used in air quality monitoring. The technique relies upon the absorption of beta radiation by solid particles extracted from air flow

in determining $PM_{10}$ and $PM_{2.5}$ concentration (USEPA, 2009). The particulate matter monitors were installed at 1.5 m above the ground. Hourly $PM_{10}$ and $PM_{2.5}$ concentration data were expressed in $\mu g\ m^{-3}$ (MEP, 2011).

Air quality index (AQI) data were obtained from nationwide air quality monitoring statistics published by the MEP, China (http://datacenter.mep.gov.cn). These AQI data were used to illustrate the influence of airborne dust versus other air pollutants on ambient air quality. In our present study, we assessed AQI based upon only particulate matter concentrations.

The AQI is calculated according to the concentration of a single air pollutant (USEPA, 2006; Wang et al., 2013), such as $PM_{10}$ and $PM_{2.5}$, according to:

$$AQI_i = \frac{AQI_u - AQI_L}{C_u - C_L} \times (C_i - C_L) + AQI_L, \tag{1}$$

Where $AQI_i$ is the index for pollutant i (i.e., $PM_{10}$, and $PM_{2.5}$), $AQI_u$ and $AQI_L$ are the upper and lower limits of the index for a specific category of air quality (i.e. excellent, moderate, slightly pollution, moderate pollution, heavy pollution and severe

pollution), $C_i$ is the observed concentration of pollutant, and $C_u$ and $C_L$ are the upper and lower concentration limits of the pollutant for a specific category of air quality. Information regarding the determination of the AQI index can be accessed



from the MEP, China (MEP, 2012a, 2012b). Based on the AQI, air quality was classified as: excellent, with AQI ≤50; moderate, with AQI 50-100; lightly pollution, with AQI 100-150; moderate pollution with AQI 150-200; heavy pollution, with AQI 200-300; and severe pollution, with AQI 300-500. Air pollution levels were classified at six levels as listed in Table 1.

Hourly ambient $PM_{10}$ concentration data were also collected at select locations across the western United States from the U.S. EPA (Environmental Protection Agency) at https://aqs.epa.gov/aqsweb/airdata/download_files.html. Data were collected from May 2-15 in an attempt to identify elevated concentrations arising from long-range transport of dust from Asia

### 2.1.2 Satellite data

MODIS Terra satellite data was collected from the U.S. National Aeronautics and Space Administration

(https://terra.nasa.gov). The Terra satellite images the entire Earth's surface every one to two days in 36 discrete spectral bands. MODIS Level 3 Deep-blue products of aerosols optical depth (AOD) were collected for analysing the spatiotemporal distribution of dust aerosols across large spatial scales (Hyer et al., 2011). This algorithm product provides comprehensive properties of aerosol optical depth, Ångström exponent, and total column optical extinction of aerosol at a wavelength of 550 nm. This wavelength has been used to quantitatively track the evolution of global dust and fine-mode anthropogenic aerosols

(Hsu et al., 2006).

CALIPSO (Cloud-Aerosol Lidar and Infrared Pathfinder Satellite Observations) was launched on 28 April, 2006 to study the roles of clouds and aerosols on climate and weather. The satellite flies in the international "A-Train" constellation for coincident Earth observations and is comprised of three instruments: the Cloud-Aerosol Lidar with Orthogonal Polarization (CALIOP Lidar), the Imaging Infrared Radiometer (IIR), and the Wide Field Camera (WFC). Passive and active remote

sensing instruments on board the CALIPSO satellite monitor aerosols and clouds 24 hours a day. CALIPSO's temporal and spatial resolution are 0.74 seconds and 333 m, respectively. We used CALIPSO aerosol optical depth (AOD) data at 532 nm of Vertical Feature Mask (VFM) level 2 version 4.10 in this study to analyse mineral dust transport across the North Pacific Ocean. This standard product of CALIPSO datasets, which corresponded to sub-micron and super-micron radius regimes, is derived at the reference wavelength. The utility of using CALIPSO products

(https://eosweb.larc.nasa.gov/project/calipso/cal_lid_l2_vfm-standard-v4-10) lies in the fact that atmospheric aerosols and clouds can be detected and classified into clean marine, dust, polluted continental, clean continental, polluted dust, smoke, etc. This robust characterization was useful in analysing the vertical distribution and variation of atmospheric components.

### 2.1.3 Meteorological data

Meteorological data, including synoptic conditions, surface wind speed, and visibility were collected for each of the

meteorological observation stations in East Asian countries from the China Meteorological Administration. Observations were taken every three hours. Dust synoptic conditions was mainly defined by visibility and subjective synoptic reports according to World Meteorological Organization (WMO) protocol. Both "present weather" and "past weather" conditions were recorded by the meteorological observer with descriptions in specified format and codes. Those codes specified at each





reporting        time       designates     the       intensity     and      duration     of      dusty      periods
(http://www.wmo.int/pages/prog/www/WMOCodes.html; Shao and Dong, 2006).

2.1.4 Lidar data

Lidar data were collected from AD-Net (http://www-lidar.nies.go.jp/AD-Net). Vertical distribution of dust in the atmosphere

was observed by Lidar at meteorological stations in Beijing, Seoul, Matsue, Osaka, and Tokyo (Fig. 2 and Table 2). The
ground-based Lidars were developed by the Japanese National Institute for Environmental Studies (NIES) and operated as
part of the Japanese NIES Lidar network and the Asian dust network (Murayama et al., 2001; Shimizu et al., 2004). Dust
particles tend to be highly non-spherical and show a high degree of depolarization (Cottle et al., 2013). Therefore, in this
study, we used non-spherical data observed by the Lidar network. Depolarization at dual-wavelength channels of 1064 nm

and 532 nm was used to identify aerosol types from the Lidar measurements (Sugimoto et al., 2003; Shimizu et al., 2004).
The laser beam was vertically oriented toward the sky and collimated with a beam expander. The beam had an output power
of 20 mJ/pulse at 1064 nm (30 mJ/pulse at 532 nm) and a pulse repetition rate of 10 Hz (Shimizu et al., 2016). The scattered
light was received by a 20 cm Schmidt Cassegrain type telescope which separated the light at 532 nm and 1064 nm
(Sugimoto et al., 2008). The measured Lidar signal was collected every 15 minutes with a vertical resolution of 30 m.

Detailed information on the calibration method and its accuracy can be found in Shimizu et al. (2004).

## 2.2 WRF-Chem model

The numerical atmospheric WRF-Chem model used in our study (version 3.7.1 available at http://ruc.noaa.gov/wrf/WG11)
has a simulation domain that covers the Asia-Pacific region. The model is run using National Center for Atmospheric
Research/National Center of Environmental Prediction (NCAR/NCEP) reanalysis meteorological input data

(http://rda.ucar.edu/datasets/ds083.2) at a horizontal resolution of $1^\circ \times 1^\circ$ and vertical resolution of 26 levels. The WRF-
Chem model included the following components: Noah land surface scheme, Yonsei University planetary boundary layer
scheme, MM5 similarity surface layer scheme, WRF single-moment 5-class microphysics scheme, and the Grell 3-D
cumulus scheme.

The WRF-Chem model simulates dust aerosol emission, transport and deposition. Simulations were carried in the study area

from 25 April to 10 May, 2017. The first week of simulation (25 April -1 May) was a spin-up period to reduce the impact of
initial conditions. Emission of dust particles from the surface is a key component in the surface exchange process (Wesely
and Hicks, 2000). Dust emission flux is also closely related to the mass of dust deposition (Whicker et al., 2014). The Dust
emission modules inside WRF-Chem include Georgia Tech/Goddard Global Ozone Chemistry Aerosol Radiation and
Transport (GOCART) (Chin et al., 2000; Ginoux et al., 2001), Model for Simulating Aerosol Interactions and Chemistry

(MOSAIC), modified GOCART, Shao (2001) scheme, Shao (2004) scheme, and Shao et al. (2011b) scheme. In this study,
we selected the dust emission scheme of Shao et al. (2011b) to apply in the WRF-Chem model as this scheme has been
widely used and tested in East Asian Gobi Desert region. The Shao et al. (2011b) dust emission scheme classified particles



into four bin sizes: 0-2.5, 2.5-5, 5-10 and 10-20 μm. The dust deposition scheme used Wesely's aerodynamic resistance model (Wesely, 1989) to simulate diffusion of particulates through the air.

### 2.3 Trajectory model

Trajectory models are widely used to identify the pathway of air parcels over complex terrain and the influence of these air parcels on ambient air quality. We simulated the pathway of air parcels using the FLEXPART model (version 9.0.2 available at http://www.flexpart.eu) which is a Lagrangian transport and dispersion model designed for calculating long-range and mesoscale dispersion of air parcels (Brioude et al. 2013). This model simulates forward in time to trace particles from source areas or backward in time to backtrack particles from given receptors. In this study, we simulated 216-hour forward-trajectories at 00, 12 and 24 UTC each day during the period May 2-10, 2017. The trajectories were simulated starting at the receptor point of potential dust emission sources (discussed in Section 3.2) from 2000 to 3000 m a.s.l (above sea level). Input data for the FLEXPART model were derived from the NCEP Global Data Assimilation System mesoscale meteorological global model. These data included 6 hour products such as temperature, precipitation, wind speed, relative humidity and geopotential height for 23 levels.

## 3 Results and discussion

### 3.1 Pervasive air pollution

Figure 1 shows an overview of the severe dust storm that developed over East Asia on 2-7 May, 2017 using data from the MODIS Terra sensor. The dust storm originated in the Mongolian Gobi Desert, Hexi Corridor, and Taklimakan Desert on 2 May, 2017. Dense yellow dust clouds moved quickly from Mongolia and north Inner Mongolia across China and into the southeast China coast, Korean peninsula, and Japan. Dense dust clouds masked the North China Plain and northeast China as these regions were not visible from space. Figure 3 mimics the movement of the dust cloud across China according to the spatiotemporal variation in hourly average $PM_{10}$ concentration at 367 Chinese environmental monitoring stations from 2-7 May, 2017. The highest hourly $PM_{10}$ dust concentration (4277 μg m$^{-3}$) was observed over Inner Mongolia at Erdos (39.59°N, 109.77°E) on 3 May, 2017. On 4 May, 2017, the dust storm severely influenced the North China Plain with elevated $PM_{10}$ concentrations (1000-2000 μg m$^{-3}$). Beginning at 4 a.m. (CST) on 4 May, air pollution gradually deteriorated in Beijing, Tianjing, and Shijiazhuang in North China Plain as well as in Changchun, Jilin, and Tongliao in Northeast China Plain. Aeolian dust migrated eastward to the Central China Plain in the lower reaches of the Yellow River and degraded air quality. Dense dust clouds continued to move east to southeast China where high $PM_{10}$ concentrations were observed on the Shandong Peninsula on 5 May, 2017. On 5 May, dust concentrations declined in Northwest China. The Changjiang River Delta region in east-central China was affected by dust on 6-7 May, 2017. In fact, $PM_{10}$ concentrations in Shanghai and Nanjing tripled on May 6 as compared with preceding days. On 6 May, 2017, the dust cloud crossed the Changjiang River Delta region and extended to the inland Jiangxi Province in South China. The spatiotemporal variation in $PM_{2.5}$



concentration at these stations is provided in Supplement S1. $PM_{2.5}$ concentrations paralleled $PM_{10}$ concentrations across the region with $PM_{2.5}/PM_{10}$ ratio from 0.3 to 0.5 (Supplement S2).

Airborne dust is one of chief pollutants influencing air quality in China (Zhang et al., 2010). Thus, the relative contribution of mineral dust to the AQI was analysed to identify the impact of airborne dust on air quality in major cities of China. We

examined the relative contribution of dust to AQI at Alxa, Erdos and Hohhot in the arid region of north China; Beijing, Tianjing, and Shijiazhuang in the North China Plain; Xi'an, Zhengzhou and Qingdao in central China; and Wuhan, Nanjing and Shanghai in southeast China (Fig. 4). The data suggest that atmospheric dust pollution decreased from north to south and from west to east (inland to coast). Degraded air quality would have affected more than 700 million Chinese people living in the path of the dust storm.

MODIS AOD, with a quality assurance confidence of 3 over land and > 0 over ocean and a maximum retrievable signal of 1.0, was used to describe the dust load in East Asia. Mahowald et al. (2017) indicated that during dust events, AOD can be used as a reliable tool to represent dust loading in the atmosphere. High AOD values were observed in Mongolian, Inner Mongolia, Hexi Corridor, east Taklimkan Desert and Northwest China Plain (Fig. 5). The MODIS AOD data indicated atmospheric mineral dust reached east Taiwan to the south and crossed the East China Sea toward the North Pacific region.

Our comparison of MODIS AOD with WRF-Chem model simulations corroborated the spatiotemporal variation in the dust cloud throughout the Asian region.

### 3.2 Distribution of maximum wind speed

Dust storms are typically caused by anticyclone conditions that result in high winds across northwest China (Shao, 2000). The magnitude of wind speed is one of the main factors influencing dust emission, transport and deposition processes (Pye,

1987; Liu et al., 2005). Dry soil particles were uplifted from the surface by aerodynamic forces under strong winds, generating dust emission. Uplift of particles or initiation of wind erosion generally occurs at wind speeds above 7 m s$^{-1}$ (Saxton et al., 2000; Sharratt et al., 2010; Liu et al., 2015). Figure 6 presented the spatial distribution of maximum wind speed above 7 m s$^{-1}$ under the observed synoptic conditions in East Asia at the meteorological stations during 2-7 May, 2017. The data show winds were sufficient to cause wind erosion in Mongolia, north China, and the Korea peninsula. Maximum

surface wind speeds above 17.2 m s$^{-1}$ (categorized meteorologically as a strong wind) were observed in the Gobi Desert of south and southeast Mongolia (e.g., Zamiin Ude, Barum-Urt, Tsgot-Ovo), and central Inner Mongolia (e.g., Erenhot, Suniteyouqi, Xilinhot, and Hails). Synoptic records indicated that visibility was reduced to < 100 m at Guaizihu (41.37°N, 102.37°E) in the Badain Jaran Desert and at Turpan (42.83°N, 89.25°E) in the north edge of Taklimkan Desert. The maximum surface wind speed during the dust storm was 23-24 m s$^{-1}$ in the Mongolian Gobi Desert and 20-22 m s$^{-1}$ in central

Inner Mongolia, indicating these likely potential source regions during the dust event. Under strong northwest and west winds, eaolian sand and dust were blown southeast-wards to Asia-Pacific regions. Atmospheric dust was, therefore, observed in over the Korean peninsula and Japan (e.g., Asahikawa, Hakodate, Oshima, Kushiro, and Wakkanai) during 6-7 May, 2017.



### 3.3 Vertical and horizontal dust dispersion

The observed temporal and vertical dust layering structures and dust mass concentration provide information for validation of dust transport simulation studies. These dust characteristics were identified by Lidar and suggest the transport and dispersion of dust originating in northwest China were influenced by strong atmospheric circulation forcing. Figure 7

illustrates Lidar observations of vertical dust profiles in Beijing, Seoul, Matsue, Osaka and Tokyo from 2 to 10 May, 2017. Airborne dust appeared to be uplifted to altitudes of 1.5-3.5 km. Dust was clearly visible in Lidar observations beginning on 2 May in Beijing and Seoul and ending in the Korean Peninsula and Japan on 9 May. The detection of dust in Beijing and Seoul on 2 May indicates the rapidity of transport of dust from northwest China. The layering structures from Lidar were identified as dust using the depolarization ratio. Unfortunately, Lidar observations in Beijing were not possible on 6 May due

to routine maintenance. In the Lidar measurement for this study, the well-mixed boundary layers in East Asia, prior to the obvious intermingling of subsiding layers from above, show depolarization ratios 150-250% higher than boundary layers in North America (Cottle et al., 2013). This indicates that dust clouds moved across the North China Plain at altitudes <1500 m. Our Lidar observations of dust being uplifted to altitudes of 3.5 km above Korea and Japan agree closely with those of Murayama et al. (2001) who observed dust being uplifted to altitudes of 3 km over Japan and 4-5 km over Korea as a result

of a wind erosion event in Northwest China and Mongolia in 1998.

Dust arising from the surface during wind erosion events may be transported along different pathways in the atmosphere due to variations in atmospheric circulation and vertical mixing within the atmosphere across events (Satake et al., 2004). To provide insight to dust flow trajectories and the impact of dust from the potential sources (discussed in Section 3.2) on regions downwind of the source, Figure 8 depicted the 9-day forward trajectories of dust particles from 2 May, 2017 (00:00

UTC) by FLEXPART model. This numerical experiment identified the location of dust emissions according to the ground observations by WMO. Simulations indicated dust particles were released from source regions with an intensity of 1000 mg m$^{-2}$ at 2000 to 3000 m a.s.l. Strong atmospheric circulation resulted in the transport of this emitted dust to the North American continent within a week. The FLEXPART model indicated a high probability that dust particles emitted during the May 2 dust storm event crossed the North Pacific Ocean and reached the coasts of the northwestern U.S. and western Canada.

### 3.4 Atmospheric dust deposition

Long range transport of dust is influenced by dry and wet depositional processes (Shao, 2006; Tanaka and Chiba, 2006; Zheng et al., 2016). Atmospheric mineral dust and subsequent deposition in the ocean is an important source of iron in high-nutrient-low-chlorophyll (HNLC) oceanic regions (Mahowald et al., 2009). The intensity of dust deposition on land or ocean can exceed 1000 g m$^{-2}$ yr$^{-1}$ (Ginoux et al., 2001; Liu et al., 2004; Zhang et al., 2010). However, the dust deposition rate over

Chinese deserts has been reported to be 70 times larger than over the North Pacific Ocean (Shao, 2000). Atmospheric dust concentrations may change rapidly during the early stages of dust transport (Uematsu et al., 1983). Few observations exist of East Asian dust deposition over the Pacific Ocean. Uematsu et al. (1983) estimated that 1.6 Tg of dust aerosols are deposited





over the North Pacific Ocean during East Asian dust storm events. Figure 9 displays the simulated deposition of dust over East Asia and North Pacific regions. As simulated by the WRF-Chem model, approximately 29.7 Tg of dust was emitted from dust sources in Mongolia and China. Subsequently, 25.7 Tg of dust was deposited over the Asia-Pacific region with 20.4 Tg of dust deposited over land and 5.3 Tg of dust deposited over the North Pacific Ocean. Simulated results further

indicate that 4 Tg of dust were suspended in the atmosphere. The amount of dust deposited over China, Mongolia, Korea peninsula, and Japan was 14.7, 4.5, 0.2 and 0.1 Tg while the dust deposition intensity in the Yellow Sea, East China Sea and Japan Sea were 1.3, 0.2 and 0.6 g m$^{-2}$, respectively. Deposition intensity is highly correlated with atmospheric dust concentration (Shao et al., 2013; Zhang et al., 2017), thus we assume that areas with high deposition also had high atmospheric concentration. We estimated that 0.9 Tg of dust was deposited over North America. Iron deposition over the

North Pacific Ocean was estimated to be 0.19 Tg assuming the dust contained 3.5% iron (Luo et al., 2005; Mahowald et al., 2017).

Transport of dust emitted from East Asian desert sources is highly dependent on atmospheric circulation (Zhang et al., 1997). The Eurasian atmospheric circulation greatly influences weather of East Asia and is primarily driven by the strength of Asian Monsoon and the Siberian High (Park et al., 2011; Shao et al., 2013). Strong winds associated with those atmospheric

circulations cause large amounts of mineral dust to be emitted into the atmosphere, and then redeposited after long-ranged transport through wet scavenging and dry settling. According the WRF-Chem model, dust emitted from East Asian Gobi Desert sources on 2 May took 3, 3.5, and 7 d to reach the Korean peninsula, Japan, and the western coast of the United State and Canada, respectively.

Gobi and sand deserts in East Asia are important sources of global atmospheric mineral dust (Ginoux et al., 2001; Shao et al.,

2013; Chen et al., 2017). Atmospheric deposition of mixed Asian dust pollutants can result in the deposition of many compounds (e.g., sulphate, nitrate, ammonium, base cations, and heavy metals) in remote areas (Carrico et al., 2003; Li et al., 2012). Figure 10 displayed the vertical profile variations over North Pacific Ocean on 7-8 May, 2017. The profiles show atmospheric mineral dust at latitudes of 35°N-50°N on May 7 and 30°N-45°N on May 8 in the North Pacific Ocean. Dust deposition in the planetary boundary layer was also detected in the western North Pacific Ocean, which is near the source of

East Asian dust (Fig. 10a).The westerly winds aloft can carry dust raised from the surface rapidly out over the Pacific Ocean in spring which is then transported eastward. The long-range transport of atmospheric constituents from East Asia not only delivers mineral dust aerosols but also carries mixed anthropogenic pollutants and nutrients to remote continents and oceans (Li et al., 2012; Lyu et al., 2017b). As indicated from Figure 10b, 10c and 10f, anthropogenic pollution was detected in dust at an altitude of 3-8 km. This observation is consistent with the previous studies for the trans-Pacific dust transport as

reported by Huebert et al. (2003), Uno et al. (2009) and Mahowald et al. (2009).

Based upon WRF-Chem simulations, dust was emitted from localized sources in North America during 2-10 May, 2017. Tanaka and Chiba (2006) and Wu et al. (2018) also suggest that dust is emitted from localized sources in North America. There was 0.7 Tg of dust emitted across Arizona, Nevada, and the Mexican Desert during 2-10 May, 2017 according to WRF-Chem simulations. Approximately 0.7 Tg of dust was deposited in across the U.S. To visualize the influence of the



North American dust sources on atmospheric dust loading, supplementary materials S3 displays the hourly $PM_{10}$ concentration for locations in California, Arizona, Nevada and Washington during 2-15 May, 2017. These results indicate dust emitted from the Mexican Desert and drylands in Arizona and Nevada significantly influenced the atmospheric environment in the southwest U.S.

**3.5 Influence on Asia-Pacific region**

Long-range transport of mineral dust aerosols occurred with high temporal and spatial variability (Mahowald et al., 2017). Cottle et al. (2013) and Hu et al. (2016) report that long range transport of Asian dust can impact the Pacific region. High latitude dust clouds were observed crossing into North America as evidenced by CALIPSO retrieval signals in Figure 10. Dust was floating at approximately 2-8 km height over the North American atmosphere on 9-10 May (Supplement S4b and

S4e), a week after the East Asian dust storm initiated.

Asian dust can be transported to the Arctic at altitudes of 3-7 km as a result of either a blocking high pressure system in the northwest Pacific Ocean or a trough-ridge configuration between East Asia and the North Pacific Ocean (Di Pierro et al., 2011). During intercontinental transport, particulate matter more likely remained in low level plumes due to less uplift and precipitation. Long-range transport of dust aerosols in the troposphere has been detected from satellite and Lidar

observations. Our results suggest a threefold increase in dust deposition over the Pacific Ocean during dust events, emphasizing the importance of dust emission sources from East Asian lands to current ambient particulate matter levels in the environment. In general, long-range transport Asian dust originated from the Gobi Desert or other sources can significantly elevate ambient particulate matter concentration and affect air quality in major cities of China, Mongolia, Korea, Japan, and far beyond.

**4 Conclusions**

The atmospheric environment of East Asia is severely affected by dust emitted from arid and semiarid regions. This study was undertaken to quantify ambient $PM_{10}$ concentration and dust deposition in East Asia as well as its impact on the Asia-Pacific region. Based upon wind field patterns, Alxa Desert in western Inner Mongolia and the Gobi Desert of Mongolia were found to be main dust sources during the May 2017 dust storm event. This event resulted in an estimated dust emission

of 29.7 Tg. Two thirds of the emitted dust was deposited over East Asia while one six of the emitted dust was deposited into the North Pacific Ocean. This dust event dominated the atmospheric environment in East Asia. Data collected at environmental monitoring stations suggested aeolian dust was a major pollutant in East Asia affecting regional air quality and a significant contributor to the global dust budget. This dust storm impacted a wide area of China and other Asian nations and reached North America within one week. The spatial distribution and temporal variability in ambient $PM_{10}$

concentration showed significant decreases along the transport trajectory and persistence on a regional scale. Dust from other sources such as the Sahara, Middle-East and central Asia may also affect the Asia-Pacific region, therefore additional



observations would be necessary to clarify the contributions from other potential dust source regions on dust transport and deposition in East Asia and North Pacific Ocean. A combination of satellite and surface observations and numerical simulation presented in this study would reduce the uncertainties in quantitative estimations of Asian dust impact.

*Competing interests.* The authors declare that they have no conflict of interest.

*Acknowledgements.* We thank NASA Goddard Space Flight Center for providing the MODIS satellite data. We gratefully thank AD-Net (http://www-lidar.nies.go.jp/AD-Net) for providing the Lidar data in this study. The CALIPSO data were obtained from the NASA Langley Atmospheric Science Data Center (ASDC). This work was supported by the National Natural Science Foundation of China (No.41730639), the Chinese Academy of Sciences (No.131965KYSB20170038), the Ministry of Science and Technology, China (No.2017FY101004), the National Natural Science Foundation of China

(No.41301655), and the Open Funds (No.LAPC-KF-2017-01) of the State Key Laboratory of Atmospheric Boundary Layer Physics and Atmospheric Chemistry, China.

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



**Tables**

Table 1. Ambient particulate matter concentration related to air pollution levels in China.

| Levels | Air Quality Index (AQI) | PM$_{10}$ concentration ($\mu$g m$^{-3}$) | PM$_{2.5}$ concentration ($\mu$g m$^{-3}$) | Air quality rating |
|---|---|---|---|---|
| 1 | 0-50 | 0-50 | 0-35 | Excellent |
| 2 | 51-100 | 50-150 | 35-75 | Moderate |
| 3 | 101-150 | 150-250 | 75-115 | Lightly pollution |
| 4 | 151-200 | 250-350 | 115-150 | Moderate pollution |
| 5 | 201-300 | 350-420 | 150-250 | Heavy pollution |
| 6 | 301-500 | >420 | >250 | Severe pollution |



**Table 2.** List of Lidar observation stations in this study.

| No. | Station | Latitude | Longitude | Location |
|---|---|---|---|---|
| 1 | Beijing | 39.97° N | 116.37° E | Institute of Atmospheric Physics, CAS |
| 2 | Seoul | 37.46° N | 126.95° E | Seoul National University |
| 3 | Matsue | 35.48° N | 133.01° E | Shimane Prefecture |
| 4 | Osaka | 34.65° N | 135.59° E | Kinki University |
| 5 | Tokyo | 35.69° N | 139.71° E | Shinjuku |




**Table 3.** Estimates of atmospheric dust emission and deposition over the Asian-Pacific region during 2-10 May, 2017.

| Earth surface | Region | Emission (Tg) | Deposition (Tg) |
|---|---|---|---|
| Continent | Mongolia | 13.3 | 4.5 |
| | China | 16.4 | 14.7 |
| | Korea peninsula | - | 0.2 |
| | Japan | - | 0.1 |
| | Canada | - | 0.2 |
| | United States | 0.7 | 0.7 |
| Ocean | Yellow Sea | - | 0.5 |
| | East China Sea | - | 0.1 |
| | Japan Sea | - | 0.7 |
| | North Pacific | - | 5.3 |
| **Total** | | 30.4 | 25.7 |





**Figures**

**Figure 1.** Terra-MODIS images showing dust outbreak across East Asia from 2 to 7 May, 2017. Red star, rectangle and circle indicate Beijing, Seoul, and Tokyo, respectively.



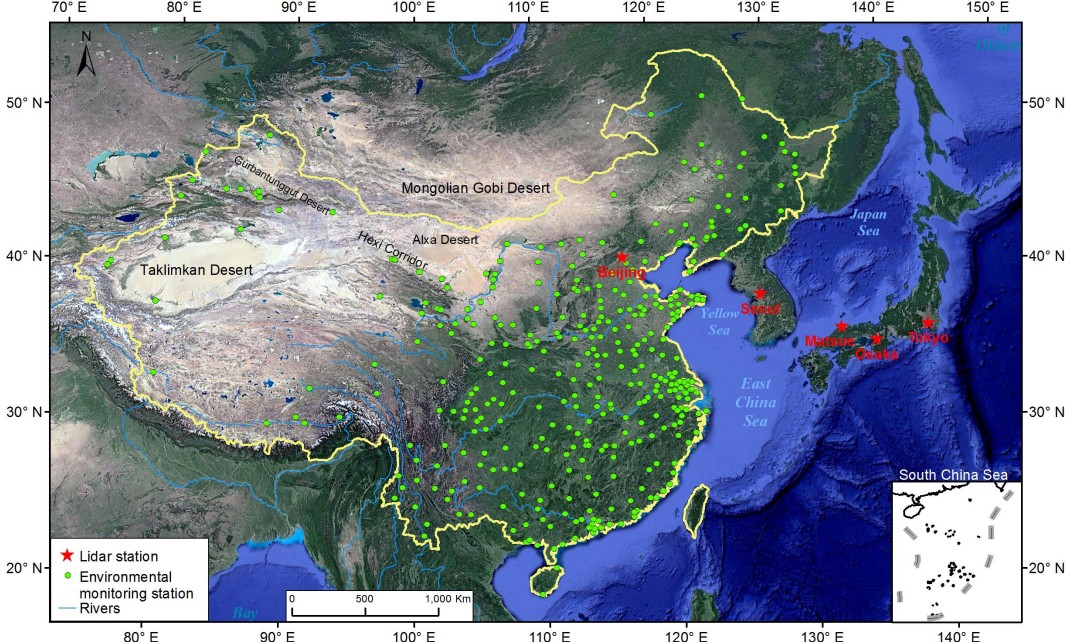

**Figure 2.** Location of environmental monitoring stations in China and Lidar monitoring stations in east Asia.





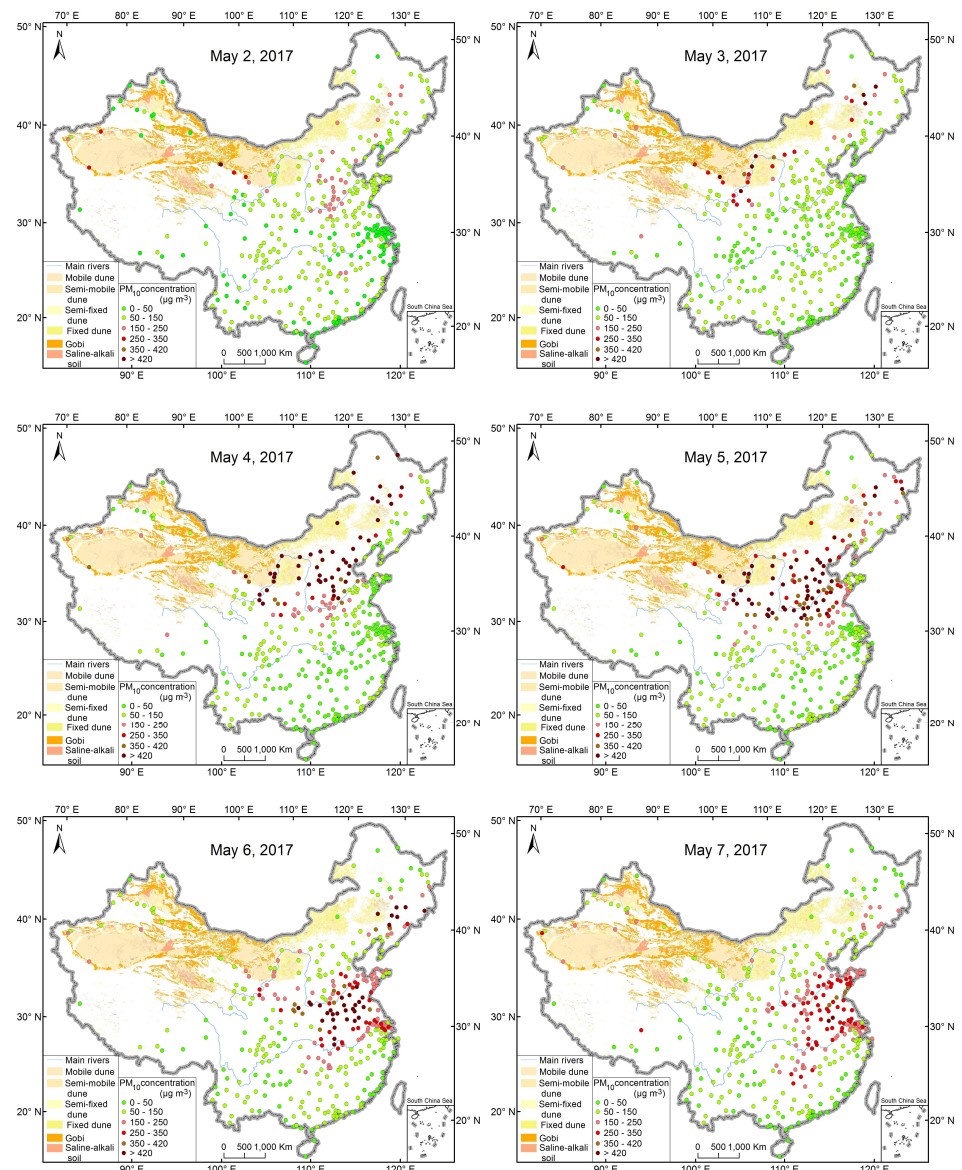

**Figure 3.** Observations of PM$_{10}$ concentrations across China during 2-7 May, 2017. PM$_{10}$ concentrations are reported as average hourly based on 24 observations during the day. Land use types are identified across the China according to Wang et al., 2005.





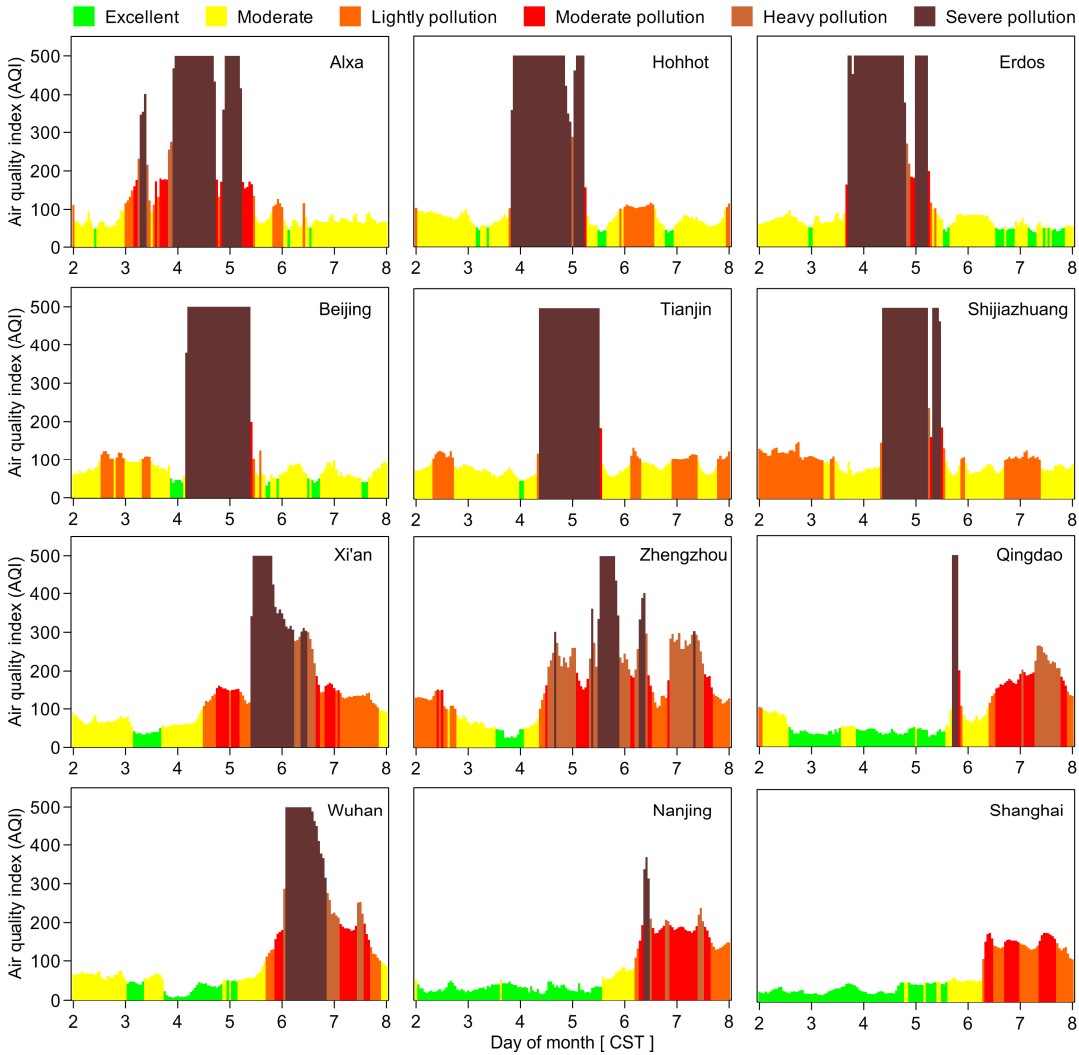

**Figure 4.** Hourly air quality index as observed for major stations in China during 2-7 May, 2017. The PM$_{10}$ pollutant contributing to the hourly AQI is identified for each station.



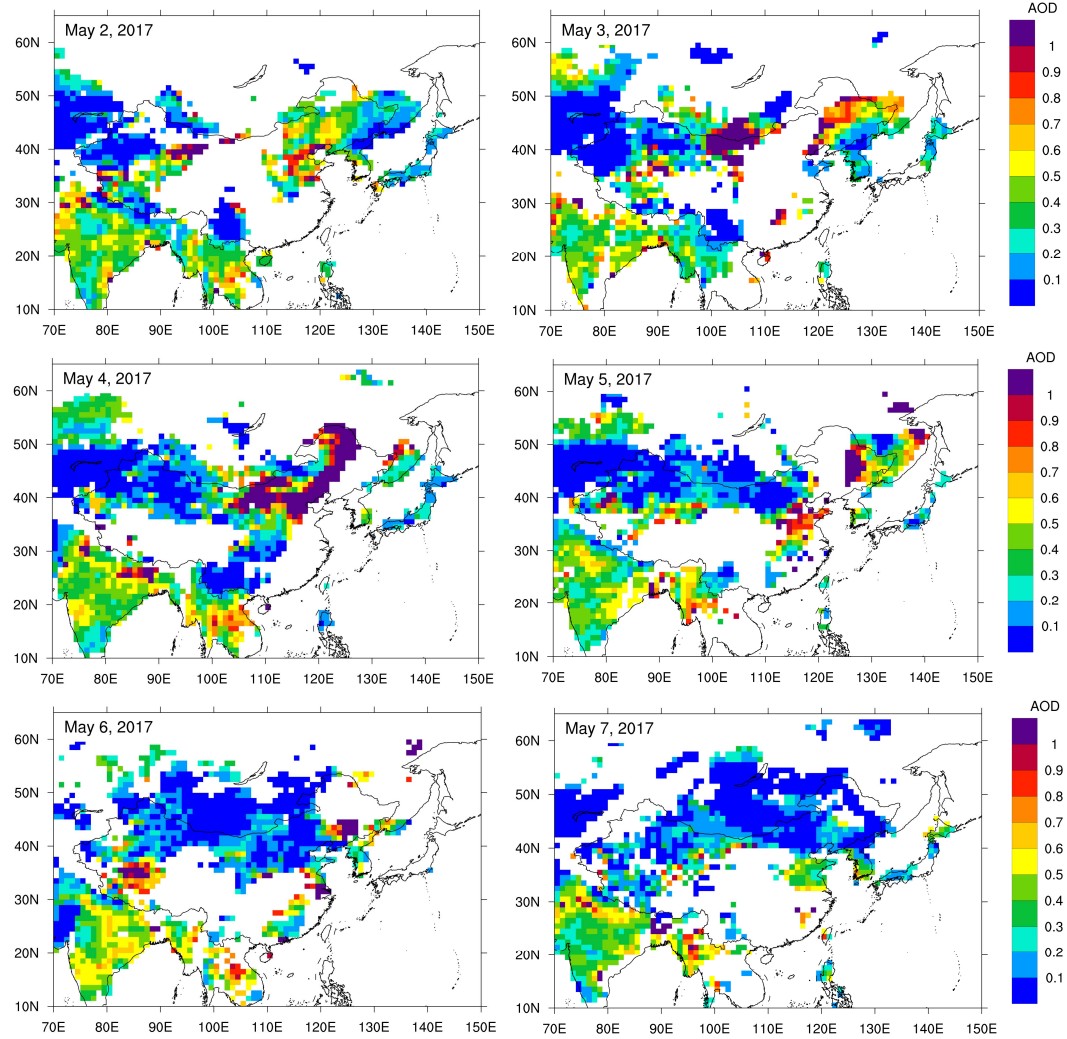

**Figure 5.** Distribution of mean aerosol optical depth (AOD) at 550 nm derived from MODIS Terra Deep-blue product in East Asia during 2-7 May, 2017.





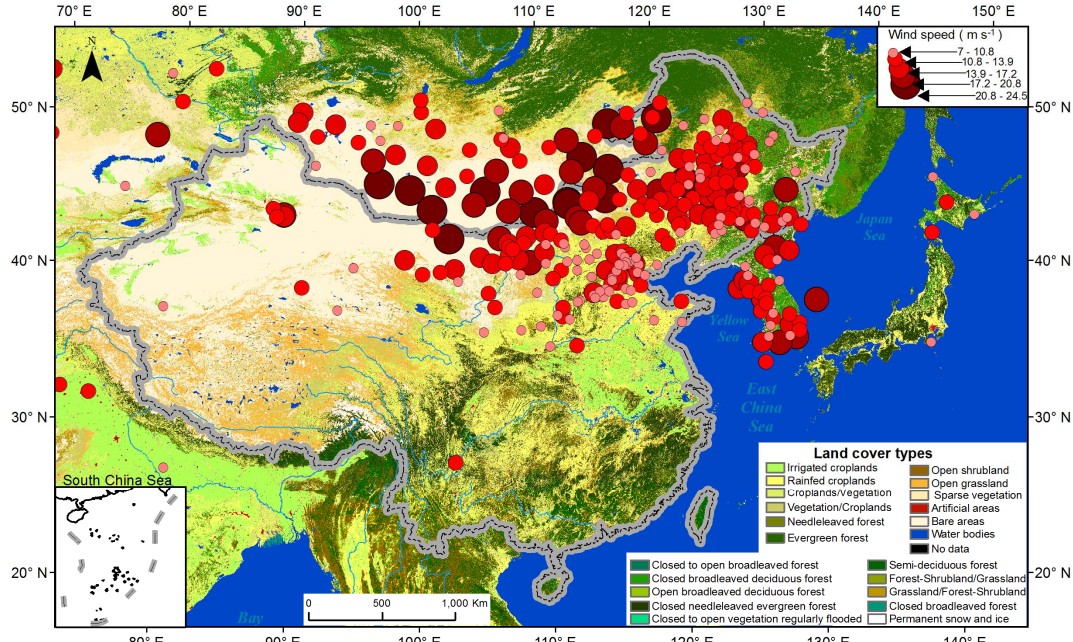

**Figure 6.** Maximum wind speed across East Asia during 2-7 May, 2017. Land cover types in East Asia are derived from global land cover products published by European Space Agency (ESA).



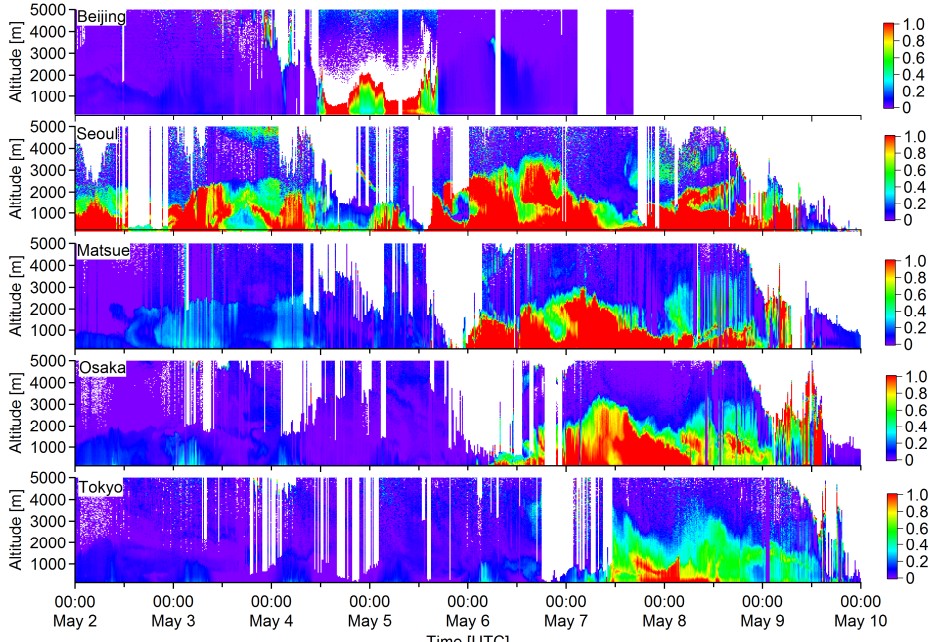

**Figure 7.** Lidar profile of dust vertical distribution in Beijing, Seoul, Matsue, Osaka and Tokyo during 2-10 May, 2017. Colors indicate depolarization ratio observed in ground Lidar.



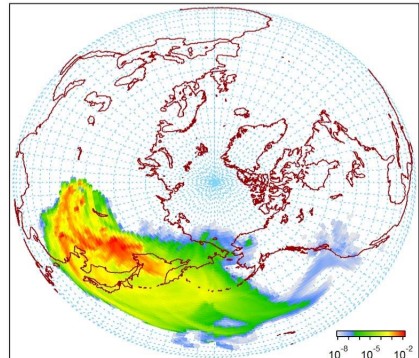

**Figure 8.** Forward trajectory analysis on dust transport (units: µg m⁻³) in free atmosphere by FLEXPART model in 216 hours from 2-10 May, 2017.





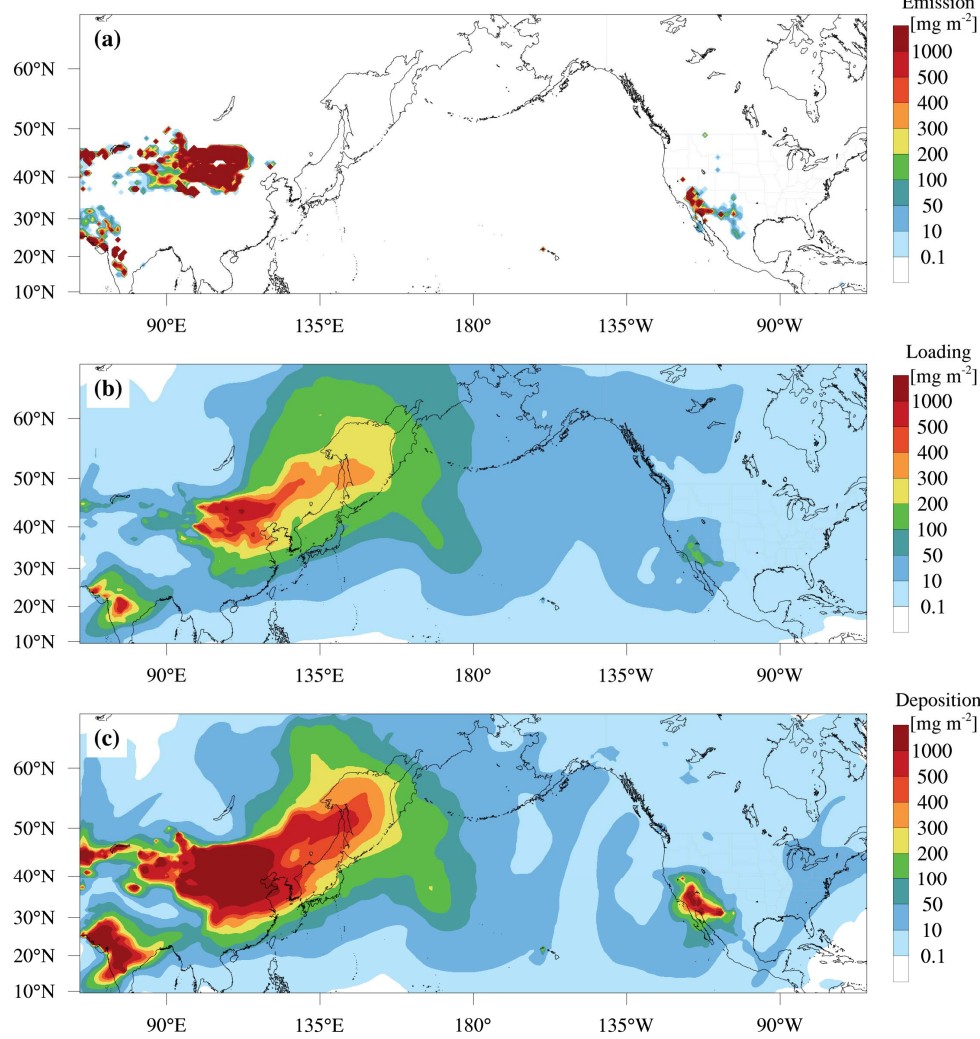

**Figure 9.** Estimation of atmospheric dust emission (a), average dust loading (b) and dust deposition (c) over Asia-Pacific region during 2-10 May, 2017.



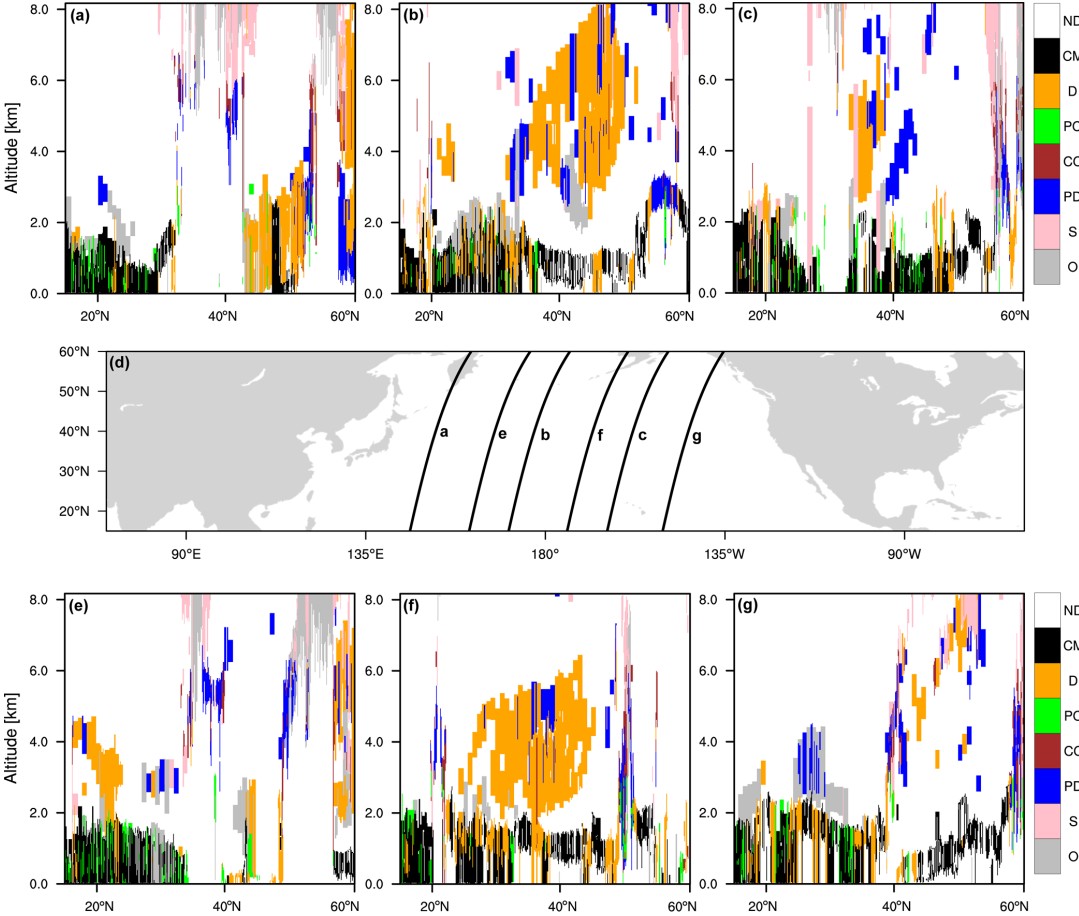

**Figure 10.** Vertical profiles of atmospheric features derived from CALIPSO satellite VFM data on 7 May (Fig.10a, 10b and 10c) and 8 May (Fig.10e, 10f, and 10g), 2017. (ND=Not determined, CM=Clean marine, D=Dust, PC=Polluted continental, CC=Clean continental, PD=Polluted dust, S=Smoke, O=Other). Each Satelllite trajectories of vertical profiles were presented in Fig. 10d.