# Peer review of "East Asian dust storm in May 2017: observations, modelling and its influence on Asia-Pacific region"

_Atmospheric Chemistry and Physics, 2018_

## Referee Comment (RC1) · Anonymous Referee #1 · 16 Apr 2018

Mineral dust is an important source of atmospheric aerosols, loess and sediments of seafloor. It is also a biogeochemical link among land, atmosphere and ocean. Unusual dust events, such as strong dust storm, are main contributor of mineral dust to the oceans, and they can influence on environment, climate, marine primary production, and atmospheric $CO_2$ at a large continental scale. Strong dust storm event that occurred in East Asia on May 2017 was a typical case to show the influence and transport of Asia dust. The authors integrated data of surface observation, remote sensing and mode simulation and analyzed quantitatively dust transport process and its influence on continental China, and northern Pacific region. The result will be very helpful to understand global dust cycle. I suggest, before it is accepted, some places should be

revised as below.

General comments: 1. In the introduction, the scientific problems and significance should be more come to the point. 2. Why data at heights of 2000-3000 m were used to analyzed the trajectories of dust transport? 3. How to exclude airborne dust in the US from its local source? 4. The transport process, such as migrating speed, impact and span on air quality at different localities, should be explained in more detail. 5. Please revise the method section to make it more concise.

Specific comments: 1. Pg 1, line 21: It's meaningless to write the number of cities directly. 2. Pg 5, line 23-24: What are those scheme used for? 3. Pg 5, line 24-28: What is the real accuracy of the simulation? 4. Pg 6, line 20-30: What indicator was used to determine the arrival of dust event in a place? 5. Pg 9, line 9-10: Some references are cited here. Is this result calculated by the authors or quoted by others? 6. Pg 9, line 1-8: I advice the authors to discuss the contribution of this event to annual dustfall in different cities of China. 7. Pg 9, line 1-8: I advice the authors to compare the magnitude of dust deposition with other dust events. 8. Fig.1: I suggest the authors explain more in detail about the dust cloud migration from MODIS image. 9. Fig.6: Please explain the maximum wind speed exactly, is it the daily maximum wind speed during the dust storm event? 10. Fig.7: Could the authors add dust vertical distribution in the source regions? 11. Fig.8: The exact meaning of loading and deposition needs to make clear.

Please also note the supplement to this comment:
https://www.atmos-chem-phys-discuss.net/acp-2018-205/acp-2018-205-RC1-supplement.pdf

---

## Referee Comment (RC2) · Anonymous Referee #2 · 10 May 2018

This paper, determined dust emission, transport, and deposition during the May 2017 Asian dust storm using environmental observations and remote sensing data along with simulation techniques, this combined approach will aid in identifying the range of transport of dust across East Asia and the North Pacific Ocean. However, there are some problems you should clearly explain and corrected before this paper is accepted. 1. Page 2, line 3, you said "This dust storm originated from the deserts of Central and East Asia, namely the Mongolian Gobi Desert, Taklimakan Desert, Hexi Corridor, and Alxa Desert (Fig. 1).".Therefore, it is recommended that you mark the Mongolian Gobi Desert, Taklimakan Desert, Hexi Corridor, and Alxa Desert position in Figure 1. 2. Page 2, line 15, you have said "Dust aerosols can be transported long distances,

even on a global scale". Therefore, I suggest you should explain in detail the path of dust aerosols transport. 3. Page 5, line 31, you have said "In this study, we selected the dust emission scheme of Shao et al.". Please explain the difference between this scheme and other programs, and further obtain the advantages of this scheme. 4. Page 6, line 26, you said "Aeolian dust migrated eastward to the Central China Plain in the lower reaches of the Yellow River and degraded air quality". It is recommended to quantify the extent of the decline in air quality by specific numerical values. 5. Page 6, line 27, you have said "Dense dust clouds continued to move east to southeast China where high PM10 concentrations were observed on the Shandong Peninsula on 5 May, 2017.". Please specify the value of PM10 at this time. 6. Page 7, line 10, you have mentioned the quality assurance confidence, please specify the calculation method of quality assurance confidence. 7. Page 8, line 29, you have mentioned "However, the dust deposition rate over Chinese deserts has been reported to be 70 times larger than over the North Pacific Ocean". Please explain how this result was obtained. 8. Page 10, line 17, you have said "In general, long-range transport Asian dust originated from the Gobi Desert or other sources can significantly elevate ambient particulate matter concentration and affect air quality in major cities of China, Mongolia, Korea, Japan, and far beyond.". Please explain how to get this result, if you get it from other articles, please list the documents that you refer to. I think the following two articles will help you: (1) Chen S., J. Huang, J. Li, R. Jia, N. Jiang, L. Kang, X. Ma, and T. Xie, 2017: Comparison of dust emissions, transport, and deposition between the Taklimakan Desert and Gobi Desert from 2007 to 2011. Science China Earth Sciences, doi: 10.1007/s11430-016-9051-0. (2) Uno, I.; Wang, Z.; Chiba, M.; Chun, Y.; Gong, S.; Hara, Y.; Jung, E.; Lee, S.; Liu, M.; Mikami, M.; Music, S.; Nickovic, S.; Satake, S.; Shao, Y.; Song, Z.; Sugimoto, N.; Tanaka, T.; Westphal, D. Dust model intercomparison (DMIP) study over Asia: Overview. Geophys Res. 2006, 111(D12), 2503-2511, DOI: 10.1029/2005JD006575. (3) Huang J P, Minnis P, Chen B, Huang Z, Liu Z, Zhao Q, Yi Y, Ayers J K. 2008. Long-range transport and vertical structure of Asian dust from CALIPSO and surface measurements during PACDEX. J Geophys Res, 113: D23212

---

## Author Response (AR1)

**Comments from anonymous Referee #1**

**General comments**

*Question 1:* In the introduction, the scientific problems and significance should be more come to the point.

*Reply:* We have revised the Introduction of the manuscript to more clearly state the problem and importance (see Page 3, Line 5-12).

*Question 2:* Why data at heights of 2000-3000 m were used to analyzed the trajectories of dust transport?

*Reply:* The trajectories from dust sources were set at 2000-3000 m based upon observations of Cottle et al. (2013a) and Cottle et al. (2013b). They reported that strong winds can cause emission and transport of dust to altitudes of 2-3 km above the ground. The text was revised to indicate the reason for using 2-3 km data to analyze trajectories (Page 5, Line 30-32).

*Question 3:* How to exclude airborne dust in the U.S. from its local source?

*Reply:* We estimated dust emissions over Asia-Pacific region using the WRF-Chem model. Simulated emissions over the western U.S. are shown in Figure 9a and Table 3. The simulations showed that the major sources of dust were Arizona and Nevada in the western U.S. as well as northern Mexico. According to CALIPSO satellite observations and trajectory analyses, Asian dust was chiefly transported across the Pacific Ocean to the U.S. at middle and high latitudes and at high altitudes. We admit that separation of airborne dust over the U.S. from local sources is difficult. However, we believe the likely source was Asian dust due to zonal transport of high altitude dust from Asia as well as seemingly little dust remaining in the atmosphere after accounting for deposition over the southwestern U.S. and northern Mexico. We have added text that briefly describes the influence of local dust sources on airborne dust over the U.S. (see Page 11, Line 1-5).

*Question 4:* The transport process, such as migrating speed, impact and span on air quality at different localities, should be explained in more detail.

*Reply:* We've added a more details on transport process and the impact of dust transport on air quality in Section 3.1. (Page 6, Line 13 to Page7, Line8). Figure 2 was also updated.

*Question 5:* Please revise the method section to make it more concisely.

*Reply:* We have further revised this section and made it concisely (Section 2, Page 3-5).

**Specific comments**

*Question 1:* Page 1, Line 21: It's meaningless to write the number of cities directly.
*Reply:* The description on the number of environmental cities and meteorological stations has been deleted in the Abstract (Page 1, Line 16).

*Question 2:* Page 5, Line 23-24: What are those scheme used for?
*Reply:* Those parametrization schemes are used for the WRF-Chem model to carry out the numerical simulation in this study. We added those schemes description for readers who's interesting to the repeatability of numerical simulation.

*Question 3:* Page 5, Line 24-28: What is the real accuracy of the simulation?
*Reply:* The modelling result on dust event on 2-7 May, 2017 with WRF-Chem is reliable. Compared with the spatiotemporal change of PM concentration, AOD and meteorological observation records, the model captured the main characteristics of this dust event. We added the explanation in Page 7, Line 23-26. Moreover the parametrization schemes such as dust emission module Shao et al. (2011b) used in this study was widely applied in East Asia. We plan to prepare another paper focusing on the modelling this dust event by WRF-Chem.

*Question 4:* Page 6, Line 20-30: What indicator was used to determine the arrival of dust event in a place?
*Reply:* The indicator that determined the arrival of dust storm event is mainly according to the meteorological records observed at each meteorological station. The WMO made a criteria on the classification of dust weather by code, which described in the Section 2.1.3 (Page 4, Line 24-28).

*Question 5:* Page 9, Line 9-10: Some references are cited here. Is this result calculated by the authors or quoted by others?
*Reply:* The result of the iron amount during the dust event was calculated by ourselves. According to the references from Luo et al. (2005) and Mahowald et al. (2017), 3.5% of total aeolian dust is the iron. Thus 5.3 Tg of dust deposited over the North Pacific Ocean included approximately 0.19 Tg iron deposition. We added this explanation in Page 9, Line 21-23.

*Question 6:* Page 9, Line 1-8: I advise the authors to discuss the contribution of this event to annual dustfall in different cities of China.
*Reply:* Currently, the environmental monitoring deposition data in major cities of China in 2017 is not available. In addition, the deposition observation during the May 2017 dust event is not systematically carried out. Thus further comparison on the contribution of this dust event to annual dustfall in Chinese cities in 2017 is difficult to discuss due to scarce observation data. Comparison of modeled data during 2-7th May 2017 to long-term annual observation data (e.g. 1981-2004) would not be appropriate because meaningful comparisons must use the same time period.

***Question 7:*** Page 9, Line 1-8: I advise the authors to compare the magnitude of dust deposition with other dust events.

***Reply:*** We've compared the dust deposition magnitude of May 2017 Asian dust event with several other severe Asian dust storm events in revised manuscript of Section 3.4 (Page 9, Line 16-19). New references have been included in this section as:

Shao, Y., Jung, E., and Leslie, L.M.: Numerical prediction of northeast Asian dust storms using an integrated wind erosion modeling system, Journal of Geophysical Research, 107(D24), 4814, doi:10.1029/2001JD001493, 2002.

Uematsu, M, Wang, Z.F., and Uno, I.: Atmospheric input of mineral dust to the western North Pacific region based on direct measurements and a regional chemical transport model, Journal of Geophysical Research, 30(6), 1342, doi:10.1029/2002GL016645, 2003.

Han, Z., Ueda, H., Matsuda, K., Zhang, R., Arao, K., Kanai, Y., and Hasome, H.: Model study on particle size segregation and deposition during Asian dust events in March 2002, Journal of Geophysical Research, 109(D19205), doi:10.1029/2004JD004920, 2004.

Li, J., Han, Z., and Zhang, R.: Model study of atmospheric particulates during dust storm period in March 2010 over East Asia, Atmospheric Environment, 45, 3954-3964, 2011.

Tan, S-C., Li, J., Che, H., Chen, B., and Wang, H.: Transport of East Asian dust storms to the marginal seas of China and the southern North Pacific in spring 2010, Atmospheric Environment, 148, 316-328, doi:10.1016/j.atmosenv.2016.10.054, 2017.

***Question 8:*** Fig.1: I suggest the authors explain more in detail about the dust cloud migration from MODIS image.

***Reply:*** We revised and added explanation in Section 3.1 (Page 6, Line 8-13).

***Question 9:*** Fig.6: Please explain the maximum wind speed exactly, is it the daily maximum wind speed during the dust storm event?

***Reply:*** The CMA provided surface wind speed of each meteorological observation station in every three hours. Therefore, the maximum wind speed in the Figure 6 is referred to the maximum wind speed at temporal resolution of three hours during the dust storm event (Page 4, Line 24-28).

***Question 10:*** Fig.7: Could the authors add dust vertical distribution in the source regions?

***Reply:*** The Figure 7 is the time series change of vertical dust distribution observed by ground-based Lidar. Currently we don't have the access of such ground-based Lidar data or product in East Asian source regions. Here we displayed vertical distribution of dust profile crossing the Gobi Desert sources regions by CALIPSO satellite observation as the following picture (see Figure. Response for Question 10).

[Figure]

**Response for Question 10.** Vertical profiles of atmospheric features derived from CALIPSO satellite VFM data on 2 May, 2017. (ND=Not determined, CM=Clean marine, D=Dust, PC=Polluted continental, CC=Clean continental, PD=Polluted dust, S=Smoke, O=Other). Each Satelllite trajectories of vertical profiles were also presented.

*Question 11:* Fig.9: The exact meaning of loading and deposition needs to make clear.
*Reply:* In Figure 9, the dust deposition is the total dust deposited onto surface of continent or ocean. The dust loading is vertical dust flux in a computational grid. Figure 9b is the hourly average dust loading. To illustrate clearly, we've changed Figure 9 into daily average with the unit of mg m$^{-2}$ d$^{-1}$. (see Page 32, Figure 9)

**Comments from anonymous Referee #2**

**Comments**

*Question 1:* Page 2, Line 3, "This dust storm originated from the deserts of Central and East Asia, namely the Mongolian Gobi Desert, Taklimakan Desert, Hexi Corridor, and Alxa Desert (Fig. 1)". Therefore, it is recommended that you mark the Mongolian Gobi Desert, Taklimakan Desert, Hexi Corridor, and Alxa Desert position in Figure 1.

*Reply:* Since Figure 1 includes all of East Asia, marking these locations in the figure would be rather difficult to see. We have instead marked the locations in Figure 2 and have revised the text to include Figure 2 at the end of the sentence "This dust storm originated from the deserts of Central and East Asia, namely the Mongolian Gobi Desert, Taklimakan Desert, Hexi Corridor, and Alxa Desert (Fig. 1 and Fig. 2)" (see Page 2, Line 3).

*Question 2:* Page 2, Line 15, "Dust aerosols can be transported long distances, even on a global scale". Therefore, I suggest you should explain in detail the path of dust aerosols transport.

*Reply:* This sentence has been revised as "Dust aerosols can be transported long distances, even on a global scale from Africa to the Americas or from Asia to North America" (Page 2, Line 15-16).

*Question 3:* Page 5, Line 31, "In this study, we selected the dust emission scheme of Shao et al.". Please explain the difference between this scheme and other programs, and further obtain the advantages of this scheme.

*Reply:* Due to the lack of cohesive wind erosion data sets, in particular dust flux data sets, none of the dust emission schemes has been rigorously tested and validated (Shao et al., 2011b). The parametrization schemes such as dust emission scheme of Shao et al. (2011b) used in this study was developed following the Shao (2001) and Shao (2004) schemes. This size-resolved dust emission scheme has been rigorously examined and validated with field measurements. The major difference between the Shao et al. (2011b) dust emission scheme and other schemes is that the dust emission formula was based on implicit physical mechanism and adjusted key parameters after field validation. It is not our intent to rank the performance of dust emission schemes. Although most of the dust emission schemes have been tested in a wind tunnel, the advantage of Shao et al. (2011b) is that it has been tested with success in East Asian arid and semi-arid regions. Notwithstanding, some dust emission schemes other than Shao et al. (2011b) are widely used in global dust studies. For consideration of East Asia dust, we selected Shao's dust emission scheme due to above reasons. In addition, Kang et al. (2011) and Wu and Lin (2014) tested several dust emission schemes in East Asia during a severe dust event, which suggested a similar viewpoint. We've added further explanation in Section 2.2 (see Page 5, Line 21-25). New references have been included in this section as follows:

Kang, J.-Y., Yoon, S.-C., Shao, Y., and Kim S.-W.: Comparison of vertical dust flux by implementing three dust emission schemes in WRF/Chem, Journal of Geophysical

Research, 116, D09202, doi:10.1029/2010JD014649, 2011.

Wu, C., and Lin, Z.: Impact of Two Different Dust Emission Schemes on the Simulation of a Severe Dust Storm in East Asia Using the WRF/Chem Model, Climatic and Environmental Research, 19(4), 419-436, doi:10.3878/j.issn.1006-9585.2013.13041, 2014.

*Question 4:* Page 6, Line 26, "Aeolian dust migrated eastward to the Central China Plain in the lower reaches of the Yellow River and degraded air quality". It is recommended to quantify the extent of the decline in air quality by specific numerical values.

*Reply:* We've added more details on the impact of dust transport on air quality in Section 3.1. (Page 6, Line 30 to Page 7, Line 4).

*Question 5:* Page 6, Line 27, "Dense dust clouds continued to move east to southeast China where high $PM_{10}$ concentrations were observed on the Shandong Peninsula on 5 May, 2017.". Please specify the value of $PM_{10}$ at this time.

*Reply:* We have added the requested information (Section 3.1, Page 7, Line 5-6).

*Question 6:* Page 7, Line 10, you have mentioned the quality assurance confidence, please specify the calculation method of quality assurance confidence.

*Reply:* The calculation method of quality assurance confidence is according to documents from the Goddard Space Flight Center, NASA, which used a MODIS-specific compositing method based on product quality assurance metrics to remove low quality pixels. These are level-2 single pixel AOD (550 nm) measurements with a QAC flag of 3 and >0 over land and Sea, respectively. The white color in Figure 5 is for missing values. We have further revised this section (see Page 7, Line 22-23). New references have been included in this section as follows:

Masuoka, E., Roy, D., Wolfe, R., Morisette, J., Sinno, S., Teague, M., Saleous, N., Devadiga, S., Justice, C., and Nickeson, J.: MODIS Land Data Products: Generation, Quality Assurance and Validation. In: Ramachandran B., Justice C., Abrams M. (eds) Land Remote Sensing and Global Environmental Change. Remote Sensing and Digital Image Processing, vol 11, Springer: New York (NY), p509-531, doi:10.1007/978-1-4419-6749-7_22, 2010.

*Question 7:* Page 8, Line 29, you have mentioned "However, the dust deposition rate over Chinese deserts has been reported to be 70 times larger than over the North Pacific Ocean". Please explain how this result was obtained.

*Reply:* This information is from Shao (2000). We have included this reference in this section (Section 3.4, Page 9, Line 10).

*Question 8:* Page 10, Line 17, "In general, long-range transport Asian dust originated from the Gobi Desert or other sources can significantly elevate ambient particulate matter concentration and affect air quality in major cities of China, Mongolia, Korea, Japan, and far beyond.". Please explain how to get this result, if you get it from other

articles, please list the documents that you refer to. I think the following two articles will help you: (1) Chen S., J. Huang, J. Li, R. Jia, N. Jiang, L. Kang, X. Ma, and T. Xie, 2017: Comparison of dust emissions, transport, and deposition between the Taklimakan Desert and Gobi Desert from 2007 to 2011. Science China Earth Sciences, doi: 10.1007/s11430-016-9051-0. (2) Uno, I.; Wang, Z.; Chiba, M.; Chun, Y.; Gong, S.; Hara, Y.; Jung, E.; Lee, S.; Liu, M.; Mikami, M.; Music, S.; Nickovic, S.; Satake, S.; Shao, Y.; Song, Z.; Sugimoto, N.; Tanaka, T.; Westphal, D. Dust model intercomparison (DMIP) study over Asia: Overview. Geophys Res. 2006, 111(D12), 2503-2511, DOI: 10.1029/2005JD006575.

*Reply:* The result was concluded from an integrated analysis of both Sections 3.3 and 3.4 as well as from previous literature. We have added an explanation and extended the analysis in this section (Section 3.5, Page 10-11). New references have been included in this section as follows:

Uno, I., Wang, Z., Chiba, M., Chun, Y.S., Gong, S.L., Hara, Y., Jung, E., Lee, S.-S., Liu, M., Mikami, M., Music, S., Nickovic, S., Satake, S., Shao, Y., Song, Z., Sugimoto, N., Tanaka, T., Westphal, D.: Dust model intercomparison (DMIP) study over Asia: Overview, Journal of Geophysical Research, 111(D12), 2503-2511, doi:10.1029/2005JD006575, 2006.

Chen, S., Huang, J., Li, J., Jia, R., Jiang, N., Kang, L., Ma, X., and Xie, T.: Comparison of dust emissions, transport, and deposition between the Taklimakan Desert and Gobi Desert from 2007 to 2011, Science China Earth Sciences, 60(7), 1338-1355, doi: 10.1007/s11430-016-9051-0, 2017.

Huang, J., Li, Y., Fu, C., Chen, F., Fu, Q., Dai, A., Shinoda, M., Ma, Z., Guo, W., Li, Z., Zhang, L., Liu, Y., Yu, H., He, Y., Xie, Y., Guan, X., Li, M., Lin, L., Wang, S., Yan, H., and Wang, G.: Dryland climate change: Recent progress and challenges, Reviews of Geophysics, 55, 719-778, doi:10.1002/2016RG000550, 2017.

[revised manuscript text omitted]
). Few studies, however, have combined the use of these techniques and monitoring data to document the fate of dust in the atmosphere and especially the range of transport of dust from its source in East Asia. Trans-Pacific transport not only has the potential to impact the North Pacific marine environment, but also air quality of communities within and downwind of the source region. The purpose of this study was therefore to determine dust emission, transport, and deposition during the May 2017 Asian dust storm using environmental observations and remote sensing data along with simulation techniques. This combined approach to understanding the fate of windblown dust will aid in identifying the range of transport of dust across East Asia and the North Pacific Ocean.

**2 Materials and methods**

**2.1 Data sources**

2.1.1 Environmental monitoringAir quality data

Environmental monitoringAir quality data were collected in mainlandfrom China and the United States during the 2-7May, 2017 dust storm event. In China, hourly PM$_{10}$ and PM$_{2.5}$ concentrationsconcentration data were measured regularly at collected from 367 environmental monitoring stations (Fig. 2) maintained by the Ministry of Environmental Protection (MEP), China. Data collected at 367 stations were used in this study and represent a spatiotemporal distribution across continental China (Fig. 2)., 2011). Ambient PM$_{10}$ and PM$_{2.5}$ concentrations ($\mu$g m$^{-3}$) were measured with an automatic beta radiation attenuation monitormonitors designed to continuously collect particulate matter and which isare widely used in air quality monitoring. The technique relies upon the absorption of beta radiation by solid particles extracted from air flow in determining PM$_{10}$ and PM$_{2.5}$ concentration (USEPA, 2009). The particulate matter monitors were installed at 1.5 m above the ground. Hourly PM$_{10}$ and PM$_{2.5}$ concentration data were expressed in $\mu$g m$^{-3}$ (MEP, 2011).

Air quality index (AQI) data were obtained from nationwide air quality monitoring statistics published by the MEP, China (http://datacenter.mep.gov.cn). These AQI data were used to illustrate the influence of airborne dust versus other air pollutants on ambient air quality. In our presentthis study, we assessed AQI based upon only particulate matter concentrations. The AQI is calculated according todependent on the concentration of a single air pollutant (USEPA, 2006; Wang et al., 2013), such as PM$_{10}$) and PM$_{2.5}$,calculated according to:

$$AQI_i = \frac{AQI_u - AQI_L}{C_u - C_L} \times (C_i - C_L) + AQI_L, \tag{1}$$

Where $AQI_i$ is the index for pollutant i (i.e., $PM_{10}$,  $PM_{2.5}$), $AQI_u$ and $AQI_L$ are the upper and lower limits of the index for a specific category of air quality (i.e. excellent, moderate, slight pollution, moderate pollution, heavy pollution and severe pollution), $C_i$ is the observed concentration of pollutant, and $C_u$ and $C_L$ are the upper and lower concentration limits of the pollutant for a specific category of air quality. Information regarding the determination of the AQI index can be accessed

5 from the MEP (MEP, 2012a, 2012b). Based on the AQI, air quality was classified as: excellent, with AQI $\leq$50; moderate, with AQI 50-100; light pollution, with AQI 100-150; moderate pollution with AQI 150-200; heavy pollution, with AQI 200-300; and severe pollution, with AQI 300-500. Air quality indices associated with particulate matter concentrations are listed in Table 1.

In the United States, hourly ambient $PM_{10}$ concentration data were  collected at 12 locations
10  maintained by the  EPA Environmental Protection Agency (EPA). Data were obtained from https://aqs.epa.gov/aqsweb/airdata/download_files.html.  for the time period May 2-15 in an attempt to identify elevated concentrations arising from long-range transport of dust from Asia.

2.1.2 Satellite data

MODIS Terra satellite data were collected from the U.S. National Aeronautics and Space Administration
15 (https://terra.nasa.gov). The Terra satellite images the entire Earth's surface every one to two days in 36 discrete spectral bands. MODIS Level 3 Deep-blue products of aerosols optical depth (AOD) were collected for analysing the spatiotemporal distribution of dust aerosols across large spatial scales (Hyer et al., 2011); products collected included comprehensive properties of aerosol optical depth, Ångström exponent, and total column optical extinction of aerosol at a wavelength of 550 nm. Extinction at a wavelength of 550 nm has been used to quantitatively track the evolution of
20 global dust and fine-mode anthropogenic aerosols (Hsu et al., 2006).

The CALIPSO (Cloud-Aerosol Lidar and Infrared Pathfinder Satellite Observations) satellite was launched on 28 April, 2006 to study the roles of clouds and aerosols on climate and weather. The satellite carries three instruments: the Cloud-Aerosol Lidar with Orthogonal Polarization (CALIOP Lidar), the Imaging Infrared Radiometer (IIR), and the Wide Field Camera (WFC). Passive
25 and active remote sensing instruments on  the  satellite continuously monitor aerosols and clouds at a temporal and spatial resolution of 0.74 seconds and 333 m, respectively. We used CALIPSO aerosol optical depth (AOD) data at 532 nm of Vertical Feature Mask (VFM) level 2 version 4.10  to analyse mineral dust transport across the North Pacific Ocean.  The utility of using CALIPSO products
30 (https://eosweb.larc.nasa.gov/project/calipso/cal_lid_l2_vfm-standard-v4-10) allows aerosols to be  classified as clean marine, dust, polluted continental, clean continental, polluted dust, and smoke.

2.1.3 Meteorological data

Meteorological data, including synoptic conditions, surface wind speed, and visibility , for more than 2000 meteorological observation stations in East  Asia were collected from the China Meteorological Administration. Observations were taken every three hours. Dust  conditions at each station were defined by visibility and subjective synoptic reports according to World Meteorological Organization (WMO) protocol. Both "present

5 weather" and "past weather" conditions were recorded by the meteorological observer with descriptions in specified format and codes. Codes were used to designate the intensity and duration of dust periods (http://www.wmo.int/pages/prog/www/WMOCodes.html; Shao and Dong, 2006).

2.1.4 Lidar data

Lidar data, which was used to examine the vertical distribution of dust in the atmosphere, were collected from AD-Net
10 (http://www-lidar.nies.go.jp/AD-Net) for meteorological stations in Beijing, Seoul, Matsue, Osaka, and Tokyo (Fig. 2 and Table 2). The ground-based Lidars were developed by the Japanese National Institute for Environmental Studies (NIES) and operated as part of the Japanese NIES Lidar network and the Asian dust network (Murayama et al., 2001; Shimizu et al., 2004). Dust particles tend to be highly non-spherical and show a high degree of depolarization (Cottle et al., 2013a), therefore we used
15 non-spherical data observed by the Lidar network. Depolarization at dual-wavelength channels of 1064 nm and 532 nm was used to identify aerosol types from the Lidar measurements (Sugimoto et al., 2003; Shimizu et al., 2004). The laser beam was vertically oriented toward the sky and collimated with a beam expander. The beam had an output power of 20 mJ/pulse at 1064 nm (30 mJ/pulse at 532 nm) and a pulse repetition rate of 10 Hz (Shimizu et al., 2016). The scattered light was received by a 20 cm Schmidt Cassegrain type telescope which separated the light at 532 nm and 1064 nm (Sugimoto et al., 2008). The
20 measured Lidar signal was collected every 15 minutes with a vertical resolution of 30 m. Detailed information on the calibration method and its accuracy can be found in Shimizu et al. (2004).

**2.2 WRF-Chem model**

The  WRF-Chem model  (version 3.7.1 available at http://ruc.noaa.gov/wrf/WG11) was used to simulate dust aerosol emission, transport and deposition across the Asia-
25 Pacific region. The model was run using National Center for Atmospheric Research/National Center of Environmental Prediction (NCAR/NCEP) reanalysis meteorological input data (http://rda.ucar.edu/datasets/ds083.2) at a horizontal resolution of $1^o \times 1^o$ and vertical resolution of 26 levels. The WRF-Chem model included the following components: Noah land surface scheme, Yonsei University planetary boundary layer scheme, MM5 similarity surface layer scheme, WRF single-moment 5-class microphysics scheme, and the Grell 3-D cumulus scheme.
30  Simulations were carried out from 25 April to 10 May, 2017 with the first week of simulation (25 April -1 May) being a spin-up period to reduce the impact of initial conditions. Emission of dust particles from the surface is a key component in the surface exchange process (Wesely and Hicks, 2000) and dust emission flux is  closely related to the mass of dust deposition (Whicker

et al., 2014).  Dust emissions is simulated by various modules in the WRF-Chem model, including the Georgia Tech/Goddard Global Ozone Chemistry Aerosol Radiation and Transport (GOCART) (Chin et al., 2000; Ginoux et al., 2001), Model for Simulating Aerosol Interactions and Chemistry (MOSAIC), modified GOCART, Shao (2001) scheme, Shao (2004) scheme, and Shao et al. (2011b) scheme. The size-resolved dust emission scheme of Shao et al. (2011b) was

5   developed based upon implicit physical mechanism following the schemes of Shao (2001) and Shao (2004). The Shao et al. (2011b) scheme has been rigorously examined and validated with field measurements, and found to perform well in simulating dust emission fluxes (Shao et al., 2011b). Therefore, in this study, we selected the dust emission scheme of Shao et al. (2011b)  in the WRF-Chem model as this scheme has been widely used and tested in East Asia Gobi Desert region (Kang et al., 2011; Wu and Lin, 2014). The Shao et al. (2011b) dust emission scheme classified particles into four bin sizes: 0-2.5,

10  2.5-5, 5-10 and 10-20 μm. Dust deposition in the WRF-Chem model was simulated  using Wesely's aerodynamic resistance model (Wesely, 1989) which accounts for diffusion of particulates through the air.

**2.3 Trajectory model**

 We used the FLEXPART model (version 9.0.2

15  available at http://www.flexpart.eu) to simulate long-range and mesoscale dispersion of air parcels over the Asia-Pacific region. This model simulates forward in time to trace particles from source areas or backward in time to backtrack particles from given receptors (Brioude et al. 2013). In this study, we simulated 216-hour forward-trajectories at 00, 12 and 24 UTC each day during the period May 2-10, 2017. The trajectories were simulated starting at the receptor point of potential dust emission sources (discussed in

20  Section 3.2) from 2000 to 3000 m a.s.l (above sea level); this range of altitude was chosen because trans-Pacific dust clouds have been observed at these altitudes (Cottle et al., 2013a; Cottle et al., 2013b). Input data for the FLEXPART model were derived from the NCEP Global Data Assimilation System mesoscale meteorological global model. These data included 6 hour products such as temperature, precipitation, wind speed, relative humidity and geopotential height for 23 levels.

**3 Results and discussion**

25  **3.1 Pervasive air pollution**

Figure 1 shows an overview of the severe dust storm that developed over East Asia on 2-7 May, 2017 using data from the MODIS Terra sensor. The dust storm originated in the Mongolian Gobi Desert, Hexi Corridor, and Taklimakan Desert on 2 May, 2017. Dense  dust clouds formed initially over southern Mongolia and western Inner Mongolia on 2-3 May, 2017, then moved quickly across north and northeast China and migrated pervasively into the southeast

30  China coast, Korean peninsula, and Japan. On 4 May, yellow dust clouds masked the North China Plain and northeast China as these regions were not visible from space. The strongest dust plumes were observed over southern Mongolia and

west Inner Mongolia on 3 May, 2017. Dust signals were accordingly captured in the Yellow Sea and East China Sea on 4-6 May, and in the north Japan Sea and sea of Okhotsk on 6-7 May. Figure 3 depicts the movement of  dust clouds across continental China according to the spatiotemporal variation in hourly average $PM_{10}$ concentration at 367 Chinese environmental monitoring stations during this dust event. The maximum observed hourly $PM_{10}$

5    concentration was above 1000 μg m$^{-3}$ in Beijing, Tianjin, Shijiazhuang, Baoding, Langfang, Xi'an and Lanzhou, and above 2000 μg m$^{-3}$ in Erdos, Hohhot, Baotou, Alxa, Shizuishan, Yan'an, Changchun and Jilin. The highest hourly $PM_{10}$ dust concentration (4277 μg m$^{-3}$) was observed over Inner Mongolia at Erdos (39.59°N, 109.77°E)  at 2 a.m. on 5 May, which was 30 times higher than the concentration observed on 2 May. Figure 3 also indicates that air quality was beginning to deteriorate in northwest China on 2 May as $PM_{10}$ concentrations were 759 μg m$^{-3}$ at Aksu and 380

10   μg m$^{-3}$ at Hotan in the Taklimakan Desert. Air quality simultaneously deteriorated along the Hexi Corridor as $PM_{10}$ concentrations were 1297 μg m$^{-3}$ in Jiuquan, 957 μg m$^{-3}$ in Zhangye, 869 μg m$^{-3}$ in Jiayuguan and 796 μg m$^{-3}$ in Jinchang. It should be noted that major cities in Northeast China Plain such as Beijing, Langfang, Shijiazhuang and Zhengzhou had level 2 or 3 air quality on 2 May due to anthropogenic pollution caused by static stability weather conditions. On 3 May, air pollution became evident in northern China, especially central Inner Mongolia and Northeast China. $PM_{10}$ concentration rose to 2500

15   μg m$^{-3}$ in Hohhot, 1540 μg m$^{-3}$ in Baotou, 898 μg m$^{-3}$ in Erdos, 1526 μg m$^{-3}$ in Shizuishan, 2403 μg m$^{-3}$ in Wuhai, 3706 μg m$^{-3}$ in Banyan Nur, 2592 μg m$^{-3}$ in Ulanqab, and 1681 μg m$^{-3}$ in Alax. In addition, $PM_{10}$ concentrations in Northeast China rose to 1634 μg m$^{-3}$ in Mudanjiang, 1056 μg m$^{-3}$ in Songyuan, and 1249 μg m$^{-3}$ in Suihua. On 4 May, the dust storm severely influenced the North China Plain as $PM_{10}$ concentrations  1000-2000 μg m$^{-3}$. Beginning at 4 a.m. (CST) on 4 May, air quality deteriorated in Beijing, Tianjing, and Shijiazhuang in the North China Plain as well

20   as in Changchun, Jilin, and Tongliao in the Northeast China Plain. Ambient $PM_{10}$ concentration in Beijing increased from 62 μg m$^{-3}$ at 3 a.m. to 491 μg m$^{-3}$ at 4 a.m. and then to 1000 μg m$^{-3}$ at 7 a.m. On 4 May, the maximum ambient $PM_{10}$ concentration at Tianjin, Shijiazhuang and Zhangjiakou was respectively 1508 μg m$^{-3}$, 1475 μg m$^{-3}$, and 2849 μg m$^{-3}$. Aeolian dust migrated south-eastward to the Central China Plain in the lower reaches of the Yellow River . Dense dust clouds continued to move east to southeast China where high $PM_{10}$ concentrations were observed on

25   the Shandong Peninsula on 5 May, 2017. On this date, the ambient $PM_{10}$ concentration in Qingdao rose from 39 μg m$^{-3}$ at 13 p.m. to 831 μg m$^{-3}$ at 17 p.m. The adverse effects of the dust storm subsided on 5 May in Northwest China as $PM_{10}$ concentrations in Aksu, Hotan, Jiayuguan, Zhangye and Jiuquan decreased respectively to 147 μg m$^{-3}$, 224 μg m$^{-3}$, 64 μg m$^{-3}$, 106 μg m$^{-3}$ and 71 μg m$^{-3}$. The Changjiang River Delta region in east-central China was affected by dust on 6-7 May, 2017. Compared with the excellent air quality ($PM_{10}$ concentrations ranged from 37-41 μg m$^{-3}$) on

30   4-5 May, $PM_{10}$ concentrations in Shanghai and Nanjing increased to 194-282 μg m$^{-3}$ on 6 May. Dust clouds crossed the Changjiang River Delta region and extended to  Jiangxi Province in South China. The $PM_{10}$ concentration in Jiujiang (Jiangxi Province) rose from 56 μg m$^{-3}$ to 225 μg m$^{-3}$ within 24 hours and peaked at 275 μg m$^{-3}$ at 2 a.m. on 7 May. The spatiotemporal variation in $PM_{2.5}$ concentration at

[revised manuscript text omitted]

dust concentrations may change rapidly during the early stages of dust transport (Uematsu et al., 1983). Few observations exist of East Asian dust deposition over the Pacific Ocean. Uematsu et al. (1983) estimated that 1.6 Tg of dust aerosols are deposited over the North Pacific Ocean during East Asian dust storm events. The annual average deposition of mineral dust over the North Pacific could be as high as 480 Tg (Uematsu et al., 2003). Figure 9 displays the simulated daily emission, loading and

5    deposition of dust over East Asia and North Pacific regions. As simulated by the WRF-Chem model, approximately 29.7 Tg of dust was emitted from dust sources in Mongolia and China. (Fig. 9a and Table 3). Subsequently, 25.7 Tg of dust was deposited over the Asia-Pacific region with 20.4 Tg of dust deposited over land and 5.3 Tg of dust deposited over the North Pacific Ocean. (Fig. 9c and Table 3). Simulated results further indicate that 4 Tg of dust were suspended in the atmosphere. The amount of dust deposited over China, Mongolia, Korea peninsula, and Japan was 14.7, 4.5, 0.2 and 0.1 Tg while the dust

10   deposition intensity in the Yellow Sea, East China Sea and Japan Sea were 1.3, 0.2 and 0.6 g m$^{-2}$, respectively. The estimated dust deposition over East Asia (20.8 Tg) during 2-7, May 2017 was of the same magnitude as deposition during the 2002 and 2010 Asian dust storms (Shao et al., 2002; Han et al., 2004; Li et al., 2011). In addition, the ratio of dust deposition intensity over the Yellow Sea to East China Sea (6.5) was close to that reported by Tan et al. (2017). Deposition intensity is highly correlated with atmospheric dust concentration (Shao et al., 2013; Zhang et al., 2017), thus we assume that areas with high

15   deposition also had high atmospheric concentration. We estimated that 0.9 Tg of dust was deposited over North America. Iron deposition over the North Pacific Ocean was estimated by ourselves to be 0.19 Tg assuming the dust contained 3.5% iron (Luo et al., 2005; Mahowald et al., 2017).

Transport of dust emitted from East Asian desert sources is highly dependent on atmospheric circulation (Zhang et al., 1997). The Eurasian atmospheric circulation greatly influences weather of East Asia and is primarily driven by the strength of Asian

20   Monsoon and the Siberian High (Park et al., 2011; Shao et al., 2013). Strong winds associated with those atmospheric circulations cause large amounts of mineral dust to be emitted into the atmosphere, and then redeposited after long-ranged transport through wet scavenging and dry settling. According the WRF-Chem model, dust emitted from East Asian Gobi Desert sources on 2 May took 3, 3.5, and 7 d to reach the Korean peninsula, Japan, and the western coast of the United State and Canada, respectively.

25   Gobi and sand deserts in East Asia are important sources of global atmospheric mineral dust (Ginoux et al., 2001; Shao et al., 2013; Chen et al., 20172017a). Atmospheric deposition of mixed Asian dust pollutants can result in the deposition of many compounds (e.g., sulphate, nitrate, ammonium, base cations, and heavy metals) in remote areas (Carrico et al., 2003; Li et al., 2012). Figure 10 displayed the vertical profile variations over North Pacific Ocean on 7-8 May, 2017. The profiles show atmospheric mineral dust at latitudes of 35°N-50°N on May 7 and 30°N-45°N on May 8 in the North Pacific Ocean. Dust

30   deposition in the planetary boundary layer was also detected in the western North Pacific Ocean, which is near the source of East Asian dust (Fig. 10a). The westerly winds aloft can carry dust raised from the surface rapidly out over the Pacific Ocean in spring which is then transported eastward. The long-range transport of atmospheric constituents from East Asia not only delivers mineral dust aerosols but also carries mixed anthropogenic pollutants and  nutrients  to remote continents and oceans (Li et al., 2012; Lyu et al., 2017b).  As indicated from Figure 10b, 10c and 10f, anthropogenic pollution was detected in dust

at an altitude of 3-8 km. This observation is consistent with the previous studies for the trans-Pacific dust transport as reported by Huebert et al. (2003), Uno et al. (2009) and Mahowald et al. (2009).

Based upon WRF-Chem simulations, dust was emitted from localized sources in North America during 2-10 May, 2017. Tanaka and Chiba (2006) and Wu et al. (2018) also suggest that dust is emitted from localized sources in North America. There was 0.7 Tg of dust emitted across Arizona, Nevada, and the Mexican Desert during 2-10 May, 2017 according to WRF-Chem simulations. Approximately 0.7 Tg of dust was deposited in across the U.S. To visualize the influence of the North American dust sources on atmospheric dust loading, supplementary materials S3 displays the hourly $PM_{10}$ concentration for locations in California, Arizona, Nevada and Washington during 2-15 May, 2017. These results indicate dust emitted from the Mexican Desert and drylands in Arizona and Nevada significantly influenced the atmospheric environment in the southwest U.S.

**3.5 Influence on Asia-Pacific region**

Long-range transport of mineral dust aerosols occur with high temporal and spatial variability (Mahowald et al., 2017). In addition, dust deposition rates are highly variable as deposition during singular dust storms can account for over 3% of the annual dust deposition flux (Liu et al., 2004; Zhang et al., 2010). For marine ecosystems, long-range transport and subsequent deposition of mineral dust can result in an influx of nutrients and thereby stimulate growth of aquatic organisms. For example, aeolian dust contains Fe which is essential to the growth of aquatic organisms such as phytoplankton (Zhuang et al., 1992; Luo et al., 2005; Mahowald et al., 2009; Mahowald et al., 2017; Tagliabue et al., 2017). Zhuang et al.(1992) proposed that Fe contained in dust may couple with anthropogenic S in the atmosphere and ocean, thereby enhancing solubility and subsequent availability to aquatic organisms. Thus, an influx of nutrients to the ocean during the May 2017 dust event would significantly influence aquatic primary productivity.

Cottle et al. (2013a) and Hu et al. (2016) reported long-range transport of Asian dust can impact the Pacific region. Indeed, high latitude dust clouds from the May 2017 event were observed crossing toward North America as evidenced by CALIPSO retrieval signals in Figure 10. Dust was observed at approximately 2-8 km height over  North America on 9-10 May (Supplement S4b and S4e), a week after initiation of the East Asian dust storm. The source of high altitude dust over North America may not be entirely from Asia, but could also be attributed to dust emissions from the southwestern U.S. and northern Mexico. Indeed, the WRF-Chem model predicted 0.7 Tg of dust emitted from these regions of North America during 2-10 May 2017 (Fig. 9a and Table 3). While we are unable to conclusively determine the source of high altitude dust over North America, we believe the likely source was Asian dust due to zonal transport of high altitude dust from Asia as well as seemingly little dust remaining in the atmosphere after accounting for deposition (dust emissions and deposition were equal as indicated in Table 3) over the southwestern U.S. and northern Mexico. Strong signals of dust aerosols were also observed at mid-latitude in the east North Pacific (Supplement Fig.S4a and S4e).

Asian dust can be transported further north to the Arctic at altitudes of 3-7 km as a result of either a blocking high pressure system in the northwest Pacific Ocean or a trough-ridge configuration between East Asia and the North Pacific Ocean (Di

Pierro et al., 2011). Mineral dust was observed throughout the vertical profile over high latitudes near the North Pole on 9 May as indicated by CALIPSO satellite data (Supplement Fig. S4a). The zonal-transported dust may mix with ice nuclei at high latitudes through microphysical nucleation process and result in cloud formation (Liu et al., 2012; Yu et al., 2012; Sand et al., 2017). Our simulations showed by the time the dust had been transported across the Pacific Ocean that >4 Tg of fine dust aerosols remained in suspension in the atmosphere (Fig. 9b and Fig. 9c). This suspended dust likely influenced the atmospheric chemical composition as well as optical properties. These anomalies of fast-rising atmospheric aerosol concentration could directly or indirectly influence the climate of temperature-sensitive regions like the Arctic (Di Pierro et al., 2011; Carslaw et al., 2013; Sand et al., 2017).

[revised manuscript text omitted]